# Reassessing the link between adiposity and head and neck cancer: a Mendelian randomization study

Fernanda Morales Berstein[1,2]*, Jasmine Khouja[1,2], Mark Gormley[1,2,3], Elmira Ebrahimi[4], Shama Virani[4], James D McKay[4], Paul Brennan[4], Tom G Richardson[1,2], Caroline L Relton[1,2,5], George Davey Smith[1,2], M Carolina Borges[1,2], Tom Dudding[1,2,3], Rebecca C Richmond[1,2]

[1]MRC Integrative Epidemiology Unit, University of Bristol, Bristol, United Kingdom; [2]Population Health Sciences, Bristol Medical School, University of Bristol, Bristol, United Kingdom; [3]University of Bristol Dental School, Bristol, United Kingdom; [4]Genomic Epidemiology Branch, International Agency for Research on Cancer, World Health Organization, Lyon, France; [5]London School of Hygiene & Tropical Medicine, London, United Kingdom

*For correspondence: dy20206@bristol.ac.uk

## eLife Assessment

This **important** study reports that higher genetically predicted adiposity is not strongly associated with the risk of head and neck cancer. The **convincing** evidence is supported by rigorous Mendelian Randomization approaches, using multiple genetic instruments and models that reduce sensitivity to pleiotropy. The work will be of interest to researchers studying cancer risk factors and genetic epidemiology.

## Abstract

**Background:** Adiposity has been associated with an increased risk of head and neck cancer (HNC). Although body mass index (BMI) has been inversely associated with HNC risk among smokers, this is likely due to confounding. Previous Mendelian randomization (MR) studies could not fully discount causality between adiposity and HNC. Hence, we aimed to revisit this using the largest genome-wide association study (GWAS) of HNC available, which has more granular data on HNC subsites.
**Methods:** We assessed the genetically predicted effects of BMI (N=806,834), waist-to-hip ratio (WHR; N=697,734) and waist circumference (N=462,166) on the risk of HNC (N=12,264 cases) and its subsites using a two-sample MR framework. We used inverse variance weighted (IVW) MR and multiple sensitivity analyses, including multivariable MR (MVMR), to explore the direct effects of the adiposity measures on HNC, while accounting for smoking behaviour (a well-known HNC risk factor).
**Results:** In univariable MR, higher genetically predicted BMI increased the risk of overall HNC (IVW OR = 1.17 per 1-SD higher BMI, 95% CI 1.02–1.34). However, the IVW effect was attenuated when smoking was included in the MVMR model (OR accounting for comprehensive smoking index = 0.96 per 1-SD higher BMI, 95% CI 0.80–1.15). Furthermore, we did not find a link between genetically predicted WHR (IVW OR = 1.05 per 1-SD higher WHR, 95% CI 0.89–1.24) or waist circumference and HNC risk (IVW OR = 1.01 per 1-SD higher waist circumference, 95% CI 0.85–1.21).
**Conclusions:** Our findings suggest that adiposity does not play a major role in HNC risk.

**Funding:** FMB was supported by a Wellcome Trust PhD studentship in Molecular, Genetic and Life-course Epidemiology (224982/Z/22/Z). RCR was supported by a Cancer Research UK grant (C18281/A29019). MCB is supported by a University of Bristol Vice Chancellor's Fellowship, the British Heart

Foundation (AA/18/1/34219) and the UK Medical Research Council (MC_UU_00032/5). GDS works within the MRC Integrative Epidemiology Unit at the University of Bristol, which is supported by the Medical Research Council (MC_UU_00032/1). CLR was supported by the Medical Research Council (MC_UU_00011/5) and by a Cancer Research UK (C18281/A29019) programme grant (the Integrative Cancer Epidemiology Programme). SV was funded by an EU Horizon 2020 grant (agreement number 825771) and NIDCR National Institutes of Dental and Craniofacial Health (R03DE030257). JK works in a unit that receives support from the University of Bristol, a Cancer Research UK grant (C18281/A29019) and the UK Medical Research Council (grant number: MC_UU_00032/7).

## Introduction

Head and neck cancer (HNC) is among the ten most common cancers in Europe, with an age standardised incidence rate of 10.3 per 100,000 person-years (*Global cancer observatory: cancer today, 2022*). Around 90% of HNCs are classed as squamous cell carcinomas of the oral cavity, pharynx, or larynx (*Curado and Hashibe, 2009*). Tobacco smoking and alcohol consumption are well-established HNC risk factors (*Hashibe et al., 2007*; *Lee et al., 2008*; *Purdue et al., 2009*; *Hashibe et al., 2009*; *Marron et al., 2010*; *IARC Working Group on the Evaluation of Carcinogenic Risks to Humans, 2004*; *Gormley et al., 2020*). High-risk human papillomavirus (HPV) infection has also been causally linked to the risk of HNC, especially oropharyngeal cancer (*Sabatini and Chiocca, 2020*; *Gillison et al., 2000*; *de Martel et al., 2020*). In contrast, the role of adiposity in the development of HNC is less well understood.

The World Cancer Research Fund/American Institute for Cancer Research (WCRF/AICR) Continuous Update Project (CUP) Expert Report published in 2018 determined higher body fatness (i.e. body mass index [BMI], waist-to-hip ratio [WHR], and waist circumference) likely increases the risk of HNC (*World Cancer Research Fund/American Institute for Cancer Research, 2018*). The CUP panel reached this conclusion even though a higher BMI has been associated with a decreased risk of HNC (*World Cancer Research Fund/American Institute for Cancer Research, 2018*), since they noted the inverse association appears to be limited to current smokers. They concluded the association between BMI and HNC risk may be biased among individuals who smoke (because smoking is a HNC risk factor associated with lower weight). It is thought that nicotine consumption could lead to appetite suppression and increased energy expenditure, which could in turn lead to weight loss (and spurious inverse associations between BMI and HNC risk [*Gaudet et al., 2015*]) among smokers (*Jo et al., 2002*). Among never smokers, BMI has been positively associated with HNC risk, in line with the evidence observed for measures of central adiposity (i.e. WHR and waist circumference; *Gaudet et al., 2015*; *Watts et al., 2024*).

However, the relationship between adiposity and smoking is complex, with evidence from Mendelian randomization (MR) studies suggesting higher adiposity increases the risk of smoking (*Taylor et al., 2019*; *Carreras-Torres et al., 2018*) while simultaneously suggesting smoking may lead to lower adiposity (*Taylor et al., 2019*; *Åsvold et al., 2014*; *Freathy et al., 2011*; *Taylor et al., 2014*; *Morris et al., 2015*). Additionally, excess adiposity, socioeconomic deprivation and (both active and passive) smoking are often strongly correlated (*Yang et al., 2008*; *Marmot and Bell, 2019*; *Marmot, 2017*; *Filippidis et al., 2016*). Thus, the positive associations between adiposity (i.e. BMI among non-smokers, WHR and waist circumference) and HNC risk may not be as unbiased as they appear.

It is important to acknowledge that previous MR studies on adiposity (i.e. BMI, WHR, waist circumference) and HNC risk were relatively small (maximum N=6034 cases) and could not fully discount causality due to limited statistical power (*Larsson and Burgess, 2021*; *Gormley et al., 2023*; *Vithayathil et al., 2021*). Therefore, the aim of this MR study was to revisit the link between adiposity and HNC risk using data from a HNC genome-wide association study (GWAS) that includes over two times the number of cases than the Genetic Associations and Mechanisms in Oncology (GAME-ON) GWAS *Lesseur et al., 2016* used by *Gormley et al., 2023* (the largest to date; N=12,619, including 6034 cases and 6585 controls) and has more granular data on HNC subsites (i.e. oral cavity, hypopharynx, oropharynx, and larynx). We also aimed to use multivariable MR (MVMR) to explore the direct effects of the adiposity measures on HNC, while accounting for smoking behaviour.

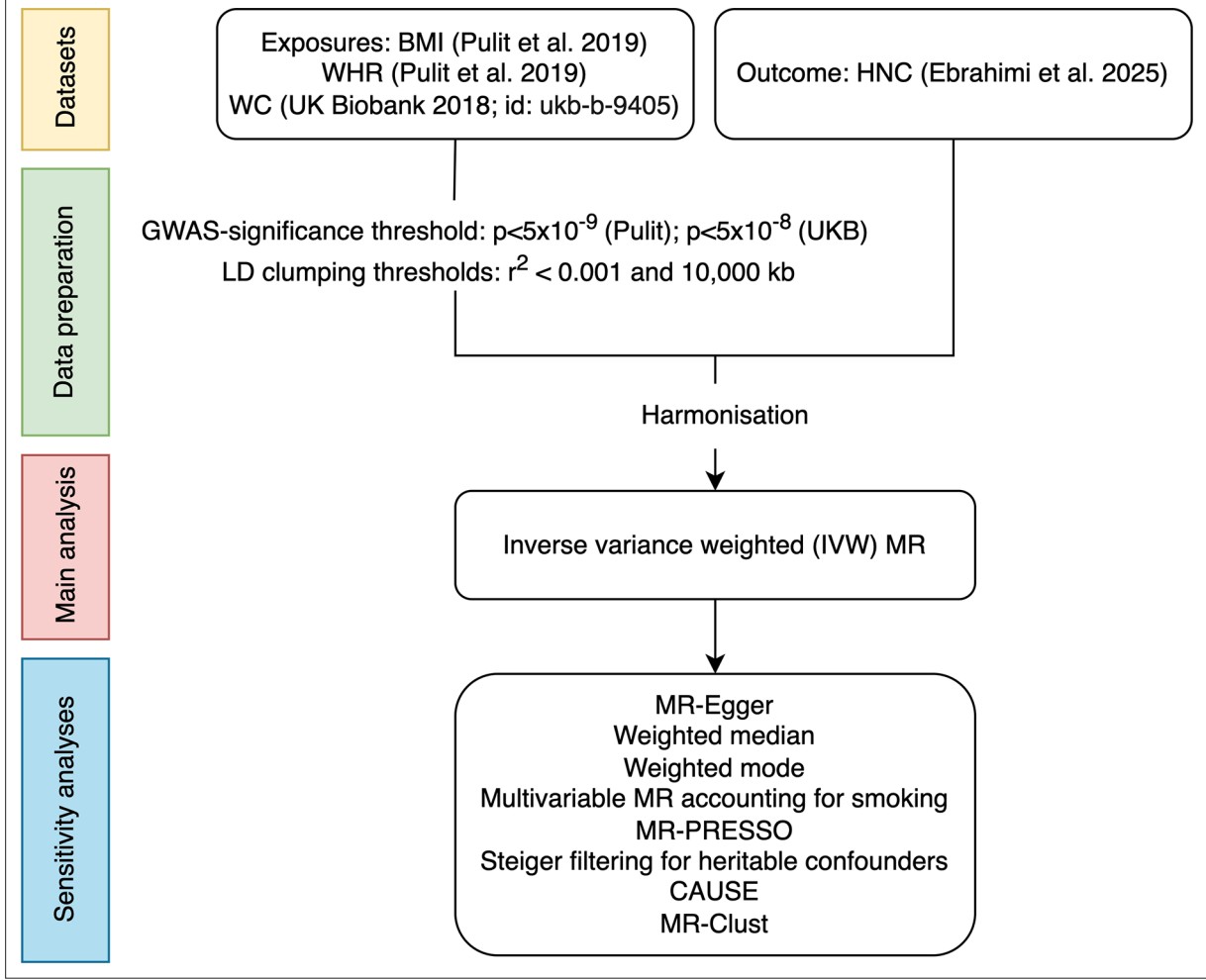

**Figure 1.** Flowchart summarising the two-sample MR framework used in this study.

## Methods

### Study design

We used a two-sample MR framework to assess the genetically predicted effects of BMI, WHR and waist circumference on the risk of HNC and its subsites (oral, laryngeal, hypopharyngeal, and oropharyngeal cancers) among individuals of European ancestry (*Figure 1*). Genetic variants associated with these adiposity traits were used as instrumental variables to estimate causal effects under the three core MR assumptions (*Davies et al., 2018*): (1) the genetic variants are strongly associated with the adiposity trait of interest (relevance assumption); (2) the distribution of the genetic variants in the population is not influenced by factors that also influence HNC risk, such as population stratification, assortative mating and dynastic effects (independence assumption); and (3) the genetic variants can only influence HNC risk via their effect on the adiposity trait of interest (exclusion restriction assumption). This work was conducted and reported according to the STROBE-MR guidelines (*Skrivankova et al., 2021*) (**STROBE-MR checklist**).

### Head and neck cancer GWAS

GWAS summary statistics for HNC were obtained from a European HEADSpAcE consortium GWAS that excluded UK Biobank participants (N=31,523, including 12,264 cases and 19,259 controls) to avoid overlapping samples across the exposure and outcome datasets. It includes the European GAME-ON data used by *Lesseur et al., 2016* and has more granular data on HNC subsites (i.e. oral cavity [N=21,269, including 3091 cases and 18,178 controls], hypopharynx [N=18,652, including

474 cases and 18,178 controls], HPV positive oropharynx [N=20,146, including 1980 cases and 18,166 controls], HPV negative oropharynx [N=19,114, including 948 cases and 18,166 controls] and larynx [N=20,668, including 2490 cases and 18,178 controls]; *Ebrahimi et al., 2024*).

HNC was defined based on the 10th revision of the International Classification of Diseases (ICD-10; *World Health Organization, 2016*). It included cancers of the oral cavity (C00.3, C00.4, C00.5, C00.6, C00.8, C00.9, C02.0–C02.9 [except C02.4], C03.0–C03.9, C04.0–C04.9, C05.0–C06.9 [except C05.1, C05.2]), the oropharynx (C01-C01.9, C02.4, C05.1, C05.2, and C09.0–C10.9), the hypopharynx (C12.0-C13.0), the larynx (C32), and overlapping or not otherwise specified sites (C14, C05.8, C02.8, C76.0; *Ebrahimi et al., 2024*).

Further detail on the HEADSpAcE GWAS has been published elsewhere (*Ebrahimi et al., 2024*). In brief, genotype data were obtained using nine different genotyping arrays. They were subsequently converted to genome build 38 for consistency across datasets. Quality control (QC) procedures were conducted by genotyping array rather than by study. Samples were excluded for the following reasons: sex mismatch (heterozygosity <0.8 for males and >0.2 for females), autosomal heterozygosity (>3 standard deviation [SD] units from the mean), missingness (>0.03), and cryptic relatedness (identity-by-decent>0.185). Single nucleotide polymorphisms (SNPs) were removed due to genotype missing-ness (>0.01), deviations from Hardy-Weinberg equilibrium (p<1e-05) and low minor allele count (<20). Imputation was performed using the TOPMed Imputation Server. Only SNPs with an imputation score $r^2$ >0.3 and a minor allele frequency (MAF) >0.005 were included in the GWAS. The analyses were conducted in PLINK using logistic regressions adjusted for sex, the top principal components and imputation batch (six in total, which account for both genotyping array and study differences).

## Genetic instruments for adiposity

GWAS summary statistics for waist circumference (N=462,166) in SD units were obtained from the UK Biobank available via the IEU OpenGWAS platform (id: ukb-b-9405). GWAS summary statistics for BMI (N=806,834) and WHR (N=697,734) in SD units were obtained from the latest Genetic Investigation of Anthropometric Traits (GIANT) consortium's GWAS meta-analysis by *Pulit et al., 2019* available at https://zenodo.org/records/1251813. The meta-analysis is the biggest to date, as it combines the meta-analysis by *Shungin et al., 2015* with UK Biobank data. The UK Biobank GWAS (*Pulit et al., 2019*) was conducted using imputed data and the BOLT-LMM software (*Loh et al., 2015*). The linear mixed models (LMMs) were solely adjusted for genotyping array. GIANT and UK Biobank data were meta-analysed (*Pulit et al., 2019*) using an inverse-weighted fixed-effect meta-analysis in METAL (*Willer et al., 2010*).

We extracted GWAS-significant SNPs for waist circumference using the standard threshold (p<5e-08). For BMI and WHR, we extracted them according to the stringent threshold recommended by *Pulit et al., 2019* to account for denser imputation data (p<5e-09). We then performed LD-clumping to select independent lead SNPs for each exposure ($r^2$=0.001, 10,000 kb). In total, 458 and 283 and 375 SNPs remained for BMI, WHR, and waist circumference, respectively.

## Data harmonisation

We extracted HNC GWAS summary statistics that corresponded to the list of SNPs selected as instruments for the exposures. Proxy SNPs ($r^2$ >0.8) were used when the instrumental SNPs were not available in the outcome datasets. Proxies were identified using the 'extract_outcome_data' function of the 'TwoSampleMR' R package and the 1000 Genomes Project European reference panel. We harmonised the exposure and outcome datasets using the 'harmonise_data' function of the 'TwoSampleMR' R package (*Hemani et al., 2018*). Positive strands were inferred using allele frequencies and ambiguous palindromic SNPs with MAFs ≥0.3 were removed. The harmonised data used in the analyses are available in *Supplementary file 1A*, *Supplementary file 1B*, *Supplementary file 1C*, *Supplementary file 1D*, *Supplementary file 1E* and *Supplementary file 1F*.

We calculated mean F-statistics and total $R^2$ values to assess the strength of our genetic instruments after data harmonisation (*Palmer et al., 2012*; *Burgess and Thompson, 2011*). Consequently, we used the total $R^2$ values to examine the statistical power in our study (*Brion et al., 2013*). However, we acknowledge the value of post-hoc power calculations is limited, since the statistical power estimated for an observed association is already reflected in the 95% confidence interval presented alongside the point estimate (*Heinsberg and Weeks, 2022*).

## Statistical analysis

### Main analyses

The multiplicative random effects inverse-variance weighted (IVW) MR approach (*Burgess et al., 2013*; the default IVW method of the 'TwoSampleMR' package [*Hemani et al., 2018*]) was used to investigate the genetically predicted effects of BMI, WHR, and waist circumference on HNC risk. We did not correct our results for multiple testing, as all our exposures are strongly correlated (*Pulit et al., 2019*; *Shungin et al., 2015*).

### Sensitivity analyses

Because the IVW method assumes all genetic variants are valid instruments (*Burgess et al., 2013*), which is unlikely the case, three pleiotropy-robust two-sample MR methods (i.e. MR-Egger [*Bowden et al., 2015*], weighted median [*Bowden et al., 2016*], and weighted mode [*Hartwig et al., 2017*]) were used in sensitivity analyses. When the magnitude and direction of effect estimates are consistent across methods that rely on different assumptions, the main findings are more convincing. As we cannot be sure about the presence and nature of horizontal pleiotropy, it is useful to compare results across methods even if they are not equally powered. We also performed tests for SNP heterogeneity (i.e. Q statistic test; *Bowden et al., 2019*) and directional horizontal pleiotropy (i.e. MR-Egger intercept test; *Bowden et al., 2015*). When directional horizontal pleiotropy was identified, we used the intercept value to evaluate the extent of the bias. The MR-PRESSO (*Verbanck et al., 2018*) method was used to identify outliers (outlier test $p<0.05$) and calculate outlier-corrected causal estimates when there was evidence of SNP heterogeneity. The MR-PRESSO distortion test was used to evaluate differences between the outlier-corrected and IVW estimates.

In addition, we ran MVMR (*Sanderson et al., 2019*) analyses to evaluate the direct effects of adiposity measures with evidence of a total effect in our main analyses. The aim of the MVMR analyses was to separate the effect of adiposity from smoking behaviour (a well-known HNC risk factor which has a complex relationship with adiposity) in the development of HNC. We obtained genetic instruments for smoking behaviour from two different sources: a smoking initiation GWAS (N=805,431 excluding 23andme) derived by the GWAS and Sequencing Consortium of Alcohol and Nicotine use (GSCAN; *Saunders et al., 2022*) and a comprehensive smoking index (CSI; a measure of lifetime smoking that captures smoking heaviness, duration and cessation) GWAS (N=462,690) conducted by *Wootton et al., 2020*. Each was separately investigated in an MVMR framework. For each smoking trait, we selected SNPs that passed the GWAS-significance and independence thresholds ($p<5e-08$, $r^2=0.001$, 10,000 kb) and combined them with the list of SNPs identified as instruments for the relevant exposure. We then performed LD-clumping across the combined list of SNPs, to then use these independent SNPs in MVMR analyses. The exposure and outcome datasets were harmonised to the same effect allele using the 'harmonise_data' function of the 'TwoSampleMR' R package (*Hemani et al., 2018*). We formatted the data using the 'format_mvmr' function of the 'MVMR' R package, calculated the conditional F-statistics for the MVMR instruments using the 'strength_mvmr' function and ran the MVMR analyses using the 'ivw_mvmr' function.

Since we used large GWAS datasets for the selection of our genetic instruments, our analyses are at an increased likelihood of being biased due to correlated horizontal pleiotropy (sometimes referred to as heritable confounding), which occurs when the genetic instruments are associated with the exposure through their effect on confounders of an exposure-outcome association (*Darrous et al., 2021*; *Morrison et al., 2020*). To mitigate this bias, we used Steiger filtering (*Hemani et al., 2017*) to remove SNPs that are more strongly associated with smoking behaviour (a confounder of the exposure-outcome association) than the exposure of interest, as proposed by *Sanderson et al., 2024*. We also used Causal Analysis using Summary Effect Estimates (CAUSE) (*Morrison et al., 2020*), another pleiotropy-robust MR method, to further investigate whether our results could be biased by correlated horizontal pleiotropy. CAUSE uses Bayesian expected log pointwise posterior predictive densities (ELPDs) to compare null, sharing, and causal models. A higher ELPD represents a better model fit, so a positive delta ELPD (where delta ELPD = ELPD model 1 - ELPD model 2) suggests model 1 fits the data better than model 2, while a negative delta ELPD suggests the opposite. If we find evidence to reject the null hypothesis that the sharing model (i.e. causal effect fixed at zero) fits the data at least as well as the causal model (i.e. causal effect can differ from zero), our findings would

**Table 1.** Data sources and instruments for other adiposity-related anthropometric measures.

| Study | Year | Data source | Trait | Unit | Download link or OpenGWAS ID |
|---|---|---|---|---|---|
| *Ried et al., 2016* | 2016 | GIANT | Body shape PC1 (overall adiposity) | SD | https://www.joelhirschhornlab.org/giant-consortium-results |
| | | | Body shape PC2 (tall and slim vs short and plump) | SD | |
| | | | Body shape PC3 (tall with small hip vs short with big hip) | SD | |
| | | | Body shape PC4 (high BMI and weight with small hip and waist vs low BMI and weight with big hip and waist) | SD | |
| *Richardson et al., 2020* | 2020 | UKB | Childhood body size | Change in body size category | 'ieu-b-5107' |
| | | | Adulthood body size | Change in body size category | 'ieu-b-5118' |
| *Martin et al., 2021* | 2021 | UKB | Metabolically favourable adiposity | SD | https://doi.org/10.2337/figshare.14555463.v1 |
| | | | Metabolically unfavourable adiposity | SD | |
| MRC-IEU (Elsworth) | 2018 | UKB | Body fat percentage | SD | 'ukb-b-8909' |
| *Leyden et al., 2022* | 2022 | GIANT +UKB | Brain tissue-specific BMI | SD | https://www.ncbi.nlm.nih.gov/pmc/articles/PMC8874216/bin/mmc2.xlsx |
| | | | Adipose tissue-specific BMI | SD | |

BMI = body mass index. GIANT = Genetic Investigation of Anthropometric Traits. N = number. PC = principal component. SD = standard deviation. SNP = single-nucleotide polymorphism. UKB, UK Biobank.

be consistent with a causal effect. Steiger filtering and CAUSE analyses were only conducted for adiposity measures with evidence of a total effect in our main IVW analyses.

Moreover, we used the MR-Clust algorithm (*Foley et al., 2021*) to find distinct SNP clusters underlying the relationship between adiposity measures with evidence of a total effect in our main analyses and HNC. The identification of substantial clusters could provide insight into potential causal mechanisms. It could also flag pleiotropic variables that are associated with SNPs in each cluster. We filtered SNPs with conditional probabilities <0.8. At least four SNPs needed to remain per cluster for a substantial cluster to be reported.

## Secondary analyses

In secondary analyses, we investigated the role of BMI, WHR and waist circumference on the risk of HNC by subsite (i.e. oral cavity, hypopharynx, HPV positive oropharynx, HPV negative oropharynx, and larynx). We used a Cochran's Q test to examine heterogeneity across HNC subsites.

We also explored the role of other adiposity-related anthropometric measures on the risk of HNC and its subsites. These anthropometric measures included: (1) four body shape principal components (*Ried et al., 2016*), (2) childhood and adulthood body size (*Richardson et al., 2020*), (3) metabolically favourable and unfavourable adiposity (*Martin et al., 2021*), (4) body fat percentage, and (5) brain and adipose tissue-specific BMI (*Leyden et al., 2022*). The data sources for these traits are summarised in *Table 1*.

**Table 2.** F-statistics and variance explained for other adiposity-related anthropometric measures.

| Trait | N SNPs before/after harmonisation | Total R² | Mean F-statistics (range) |
|---|---|---|---|
| Body shape PC1 (overall adiposity) | 29/28 | 16% | 54 (28–302) |
| Body shape PC2 (tall and slim vs short and plump) | 84/81 | 3.4% | 54 (30–211) |
| Body shape PC3 (tall with small hip vs short with big hip) | 28/27 | 0.9% | 41 (30–82) |
| Body shape PC4 (high BMI and weight with small hip and waist vs low BMI and weight with big hip and waist) | 10/10 | 24.7% | 42 (30–98) |
| Childhood body size | 206/198 | 3.4% | 78 (28–1102) |
| Adulthood body size | 339/324 | 4.2% | 59 (30–1109) |
| Metabolically favourable adiposity | 34/31 | 0.4% | 64 (25–400) |
| Metabolically unfavourable adiposity | 29/27 | 0.8% | 131 (25–400) |
| Body fat percentage | 377/360 | 4.7% | 59 (30–682) |
| Brain tissue-specific BMI | 140/133 | 1.2% | 61 (29–270) |
| Adipose tissue-specific BMI | 86/81 | 0.7% | 63 (30–270) |

BMI = body mass index. N = number. PC = principal component. SNP, single-nucleotide polymorphism.

## Statistical software

We completed all MR analyses using R software version 4.4.0 and the 'TwoSampleMR' v0.6.3, 'MRPRESSO' v1.0, 'MVMR' v0.4, 'cause' v1.2.0 and 'mrclust' v0.1.0 R packages. The 'ggplot2' v3.5.1 and 'ggforestplot' v0.1.0 R packages were used to create forest plots. The code used to run the MR analyses is available at http://github.com/fernandam93/adiposity_HNC_MR.

## Results

### Genetic instruments for BMI, WHR, and waist circumference

After data harmonisation and the removal of ambiguous palindromic SNPs, 442 genetic variants remained as instruments for BMI, while 267 remained for WHR and 353 for waist circumference (*Supplementary file 1A*). The mean F-statistic for BMI was 77 (range 33–844) and the total variance explained was 4.8%. For WHR, the mean F-statistic was 73 (range 33–820) and the total variance explained was 3.1%. For waist circumference, the mean F-statistic was 58 (range 30–940) and the total variance explained was 4.4%.

Using the BMI genetic instruments (total $R^2$=4.8%) and an α of 0.05, we had 80% statistical power to detect an OR as small as 1.16 for HNC risk (*Appendix 1—figure 1*). For WHR (total $R^2$=3.1%) and waist circumference (total $R^2$=4.4%), we could detect odds ratios (ORs) as small as 1.20 and 1.17, respectively. This is an improvement in terms of statistical power compared to the GAME-ON analysis published by *Gormley et al., 2023*, for which there was 80% power to detect an OR as small as 1.26 using the same BMI genetic instruments (*Appendix 1—figure 2*).

The F-statistics and $R^2$ values for the other adiposity-related anthropometric measures have been summarised in *Table 2*.

### Genetically predicted effects of BMI, WHR, and waist circumference on HNC risk

In univariable MR, higher genetically predicted BMI increased the risk of overall HNC (IVW OR = 1.17 per 1 standard deviation [1-SD] higher BMI, 95% CI 1.02–1.34, p=0.03), with no heterogeneity across subsites (Q p=0.78; *Figure 2*, *Appendix 1—figure 3* and *Supplementary file 1G*). However,

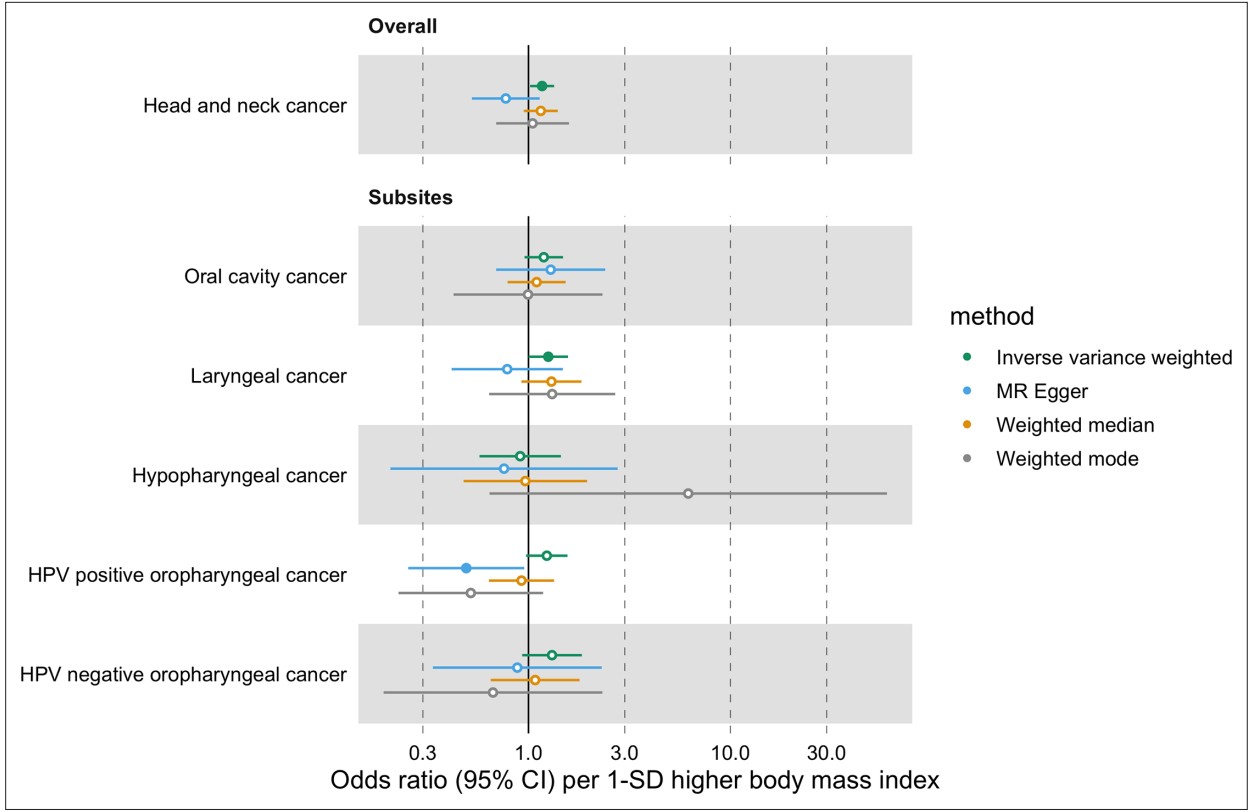

**Figure 2.** Forest plot for the genetically predicted effects of body mass index on the risk of head and neck cancer and its subsites.

the positive relationship between genetically predicted BMI and HNC risk was not consistent across the MR-Egger, weighted median and weighted mode analyses, with point estimates in opposing directions and confidence intervals including the null. The Q statistic and MR-Egger intercept tests suggested that there was heterogeneity across individual SNP estimates (Q=609, p<0.001) and a minor degree of unbalanced horizontal pleiotropy (intercept = 0.007, p=0.03) that could have biased the main IVW results (*Supplementary file 1H* and *Supplementary file 1I*). Although the MR-PRESSO analysis identified two outliers (i.e. rs11611246 and rs9603697), the distortion test suggested the outlier-corrected estimate (outlier-corrected IVW OR = 1.14 per 1-SD higher BMI, 95% CI 1.00–1.30, p=0.05) was not statistically different to the main IVW estimate (p=0.94; *Supplementary file 1J*).

Furthermore, we did not find a link between genetically predicted WHR and HNC risk (IVW OR = 1.05 per 1-SD higher WHR, 95% CI 0.89–1.24, p=0.53) and there was no heterogeneity across subsites (Q p=0.15; *Figure 3*, *Appendix 1—figure 4* and *Supplementary file 1G*). MR-Egger, weighted median and weighted mode results were consistent with a null effect. The Q statistic and MR-Egger intercept tests suggested that there was some evidence of SNP heterogeneity (Q=332, p=0.004) and unbalanced horizontal pleiotropy (intercept = 0.008, p=0.03; *Supplementary file 1H* and *Supplementary file 1I*). The MR-PRESSO analysis did not identify any significant outliers (*Supplementary file 1J*).

Similarly, we did not find a genetically predicted effect of waist circumference on HNC risk (IVW OR = 1.01 per 1-SD higher waist circumference, 95% CI 0.85–1.21, p=0.87) or evidence of heterogeneity across subsites (Q p=0.87; *Figure 4*, *Appendix 1—figure 5* and *Supplementary file 1G*). The MR-Egger, weighted median, and weighted mode consistently suggested the absence of an effect of waist circumference on HNC risk. The Q statistic and MR-Egger intercept tests suggested SNP heterogeneity (Q=563, p<0.001) but no unbalanced horizontal pleiotropy (intercept = −0.002, p=0.68; *Supplementary file 1H and I*). The MR-PRESSO analysis identified four outliers (i.e. rs1229984, rs1336486, rs17446091, and rs55726687) but the distortion test suggested the outlier-corrected estimate (outlier-corrected IVW OR = 0.98 per 1-SD higher waist circumference, 95% CI 0.84–1.15, p=0.82) was not statistically different to the main IVW estimate (p=0.12; *Supplementary file 1J*).

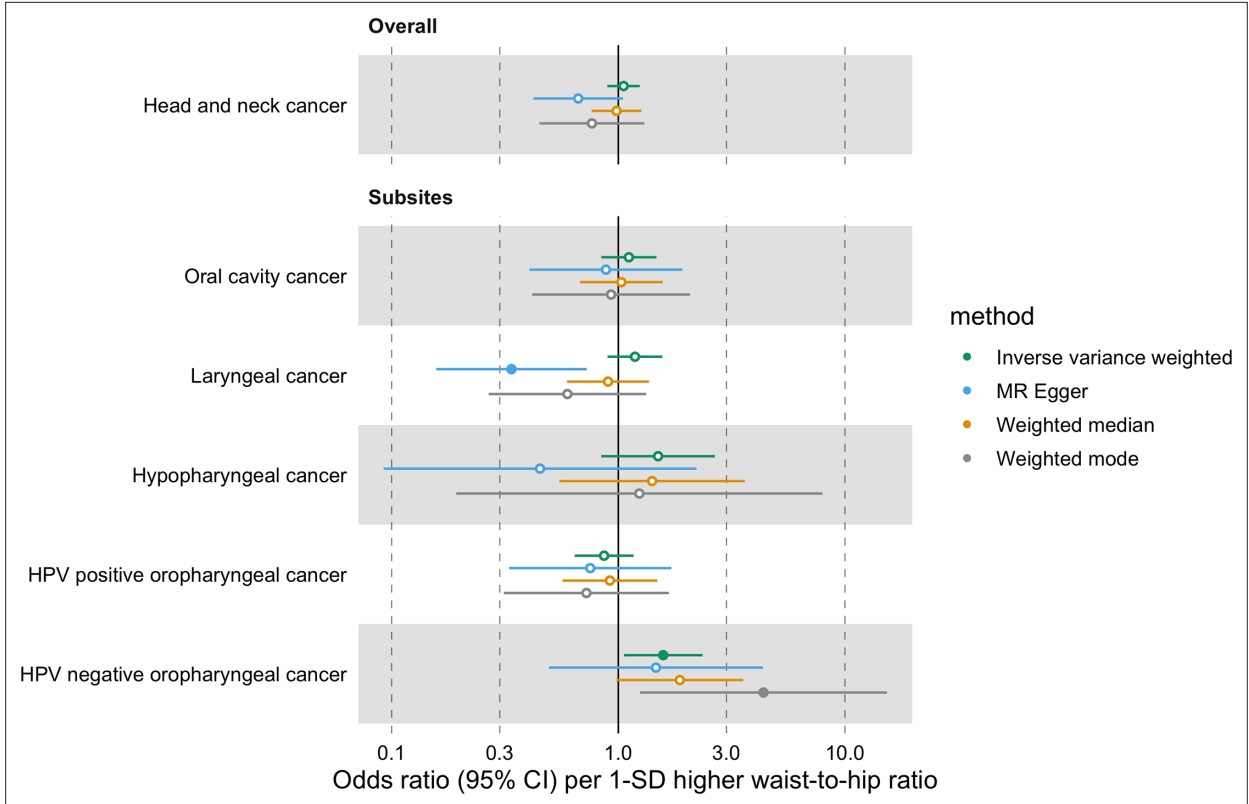

**Figure 3.** Forest plot for the genetically predicted effects of waist-to-hip ratio on the risk of head and neck cancer and its subsites.

## MVMR estimates for BMI on HNC risk after accounting for smoking behaviour

In univariable IVW MR, both CSI and SI were linked to an increased risk of HNC (CSI OR = 4.47 per 1-SD higher CSI, 95% CI 3.31–6.03, p<0.001; SI OR = 2.07 per 1-SD higher SI 95% CI 1.60–2.68, p<0.001; note in *Supplementary file 1K*).

The effect of BMI on HNC risk was attenuated when smoking behaviour was included in the MVMR model (OR accounting for CSI = 0.93 per 1-SD higher BMI, 95% CI 0.78–1.12, p=0.47; OR accounting for smoking initiation = 1.09 per 1-SD higher BMI, 95% CI 0.88–1.34, p=0.43; *Figures 5 and 6* and *Supplementary file 1K*). Genetically predicted smoking behaviour increased the risk of HNC even after accounting for BMI (CSI OR accounting for BMI = 4.25 per 1-SD higher CSI, 95% CI 3.18–5.67, p<0.001; SI OR accounting for BMI = 2.10 per 1-SD higher SI, 95% CI 1.61–2.73, p<0.001). The conditional F-statistics for the BMI estimates were 30.5 and 30.3 in the CSI and smoking initiation analyses, respectively (*Supplementary file 1K*). They were slightly lower for the smoking behaviour estimates conditioning on BMI (13.4 and 19.5 in the CSI and smoking initiation analyses, respectively).

## MR estimate for BMI on HNC risk after Steiger filtering SNPs more strongly associated with smoking behaviour than BMI

After removing six SNPs (i.e. rs10002111, rs2503185, rs264941, rs10858334, rs225882, rs2273175) that were more strongly associated with smoking behaviour (i.e. CSI or smoking initiation) than BMI, the genetically predicted effect of BMI on HNC risk slightly attenuated towards the null (Steiger filtered IVW OR = 1.14 per 1-SD higher BMI, 95% CI 1.00–1.31, p=0.05) (*Supplementary file 1L*).

## CAUSE estimate for BMI on HNC risk

We did not find evidence against bias due to correlated pleiotropy, since the causal model did not fit the data much better than the sharing model (CAUSE OR 1.12 per 1-SD higher BMI, 95% credible interval 0.93–1.34, delta ELPD for sharing vs causal = −0.07, p=0.47; *Appendix 1—figure 6*).

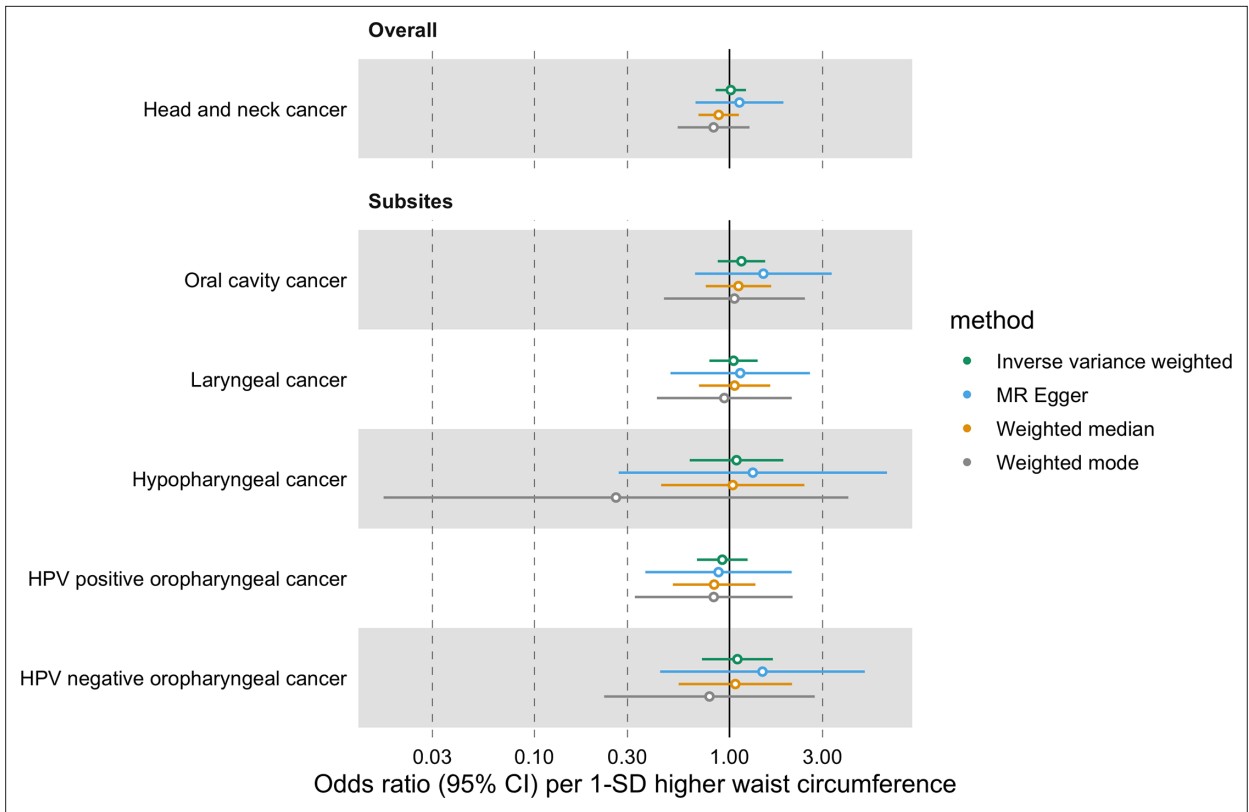

**Figure 4.** Forest plot for the genetically predicted effects of waist circumference on the risk of head and neck cancer and its subsites.

Interestingly, neither the sharing nor the causal model fitted the data much better than the null model (delta ELPD for null vs sharing = −0.39, p=0.36; and delta ELPD for null vs causal = −0.46, p=0.41).

## MR-Clust estimates for the relationship between BMI and HNC risk

After filtering SNPs with conditional probabilities <0.8 and clusters with fewer than four SNPs (e.g. cluster 1, as only three of 17 SNPs remained after probability filtering), only a null cluster including 372 SNPs (424 before filtering) remained in the MR-Clust output for BMI and HNC risk (*Appendix 1—figure 7* and *Supplementary file 1M*). Hence, the MR-Clust analysis did not reveal any mechanistic pathways underlying the effect observed.

## Genetically predicted effects of other adiposity-related anthropometric measures on HNC risk

We did not find consistent evidence of genetically predicted effects of other anthropometric measures on HNC risk (*Appendix 1—figure 8*, *Appendix 1—figure 9*, *Appendix 1—figure 10*, *Appendix 1—figure 11*, *Appendix 1—figure 12* and *Supplementary file 1N*). The IVW estimate for PC2 capturing a combination of taller height and slimmer waist suggested this body shape decreased HNC risk (OR = 0.86, 95% CI 0.75–0.99, p=0.04; *Appendix 1—figure 8b*). Similarly, the IVW estimate for PC3 capturing a combination of taller height and narrower hips suggested this body shape also reduced HNC risk (OR = 0.73, 95% CI 0.55–0.97, p=0.03; *Appendix 1—figure 8c*). However, these inverse relationships were not consistent with results obtained using pleiotropy-robust methods (i.e. MR-Egger, weighted median, and weighted mode).

## Discussion

In this MR study, we reaffirmed that there is no clear evidence of a genetically predicted effect of adiposity (i.e. BMI, WHR, and waist circumference) or related anthropometric measures on the risk of HNC or its subsites. Although we found higher genetically predicted BMI increased the risk of

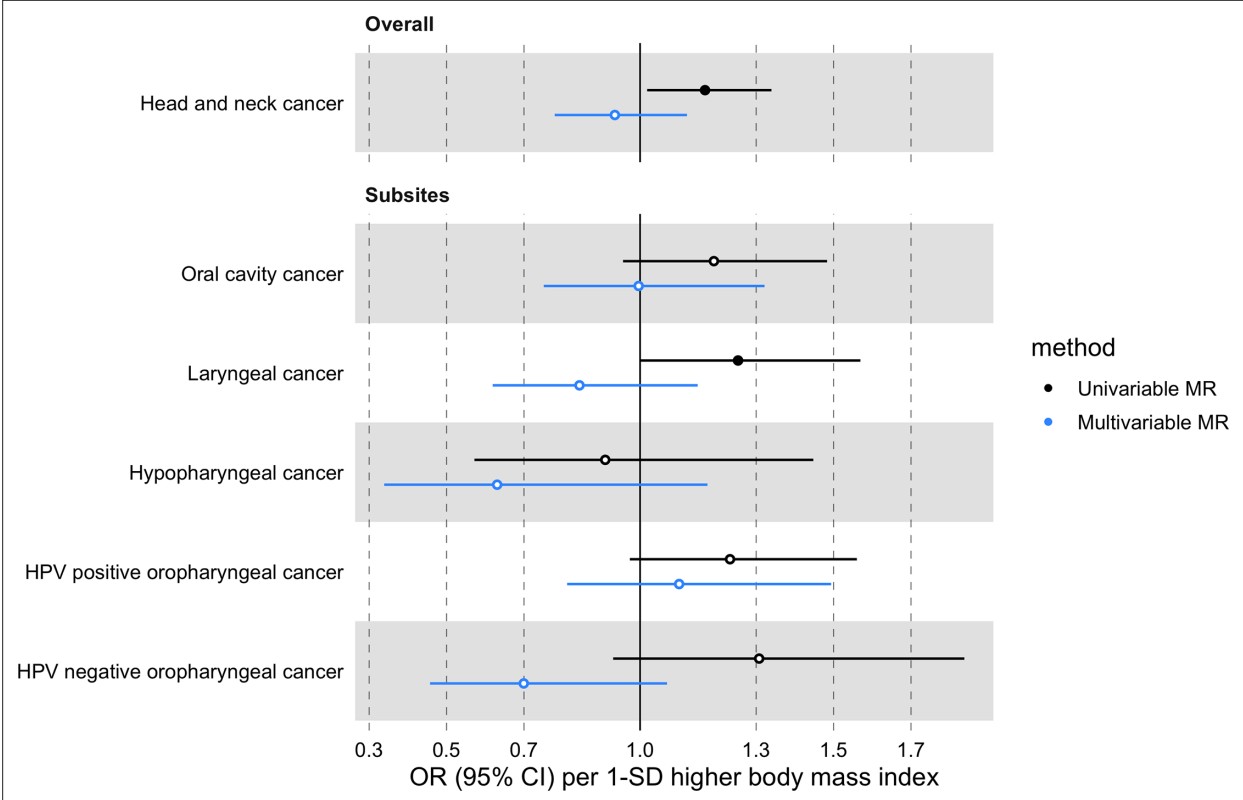

**Figure 5.** Forest plot for the genetically predicted effects of BMI on the risk of HNC and its subsites, before (univariable-black) and after (multivariable-blue) accounting for comprehensive smoking index (CSI).

overall HNC in the main univariable MR analysis, this was not consistent across the sensitivity analyses. Notably, the MVMR results suggested the main analysis may have been biased by smoking (and/or related traits), as the effect disappeared after accounting for smoking behaviour. The results obtained after Steiger filtering SNPs more strongly associated with smoking behaviour than BMI suggested correlated pleiotropy may have been partly biasing the BMI-HNC estimate. CAUSE, which is more robust to correlated pleiotropy than the IVW method, further supported this hypothesis.

Previous MR studies suggest adiposity does not influence HNC risk (*Larsson and Burgess, 2021*; *Gormley et al., 2023*; *Vithayathil et al., 2021*). *Gormley et al., 2023* did not find a genetically predicted effect of adiposity on combined oral and oropharyngeal cancer when investigating either BMI (OR = 0.89 per 1-SD, 95% CI 0.72–1.09, p=0.26), WHR (OR = 0.98 per 1-SD, 95% CI 0.74–1.29, p=0.88) or waist circumference (OR = 0.73 per 1-SD, 95% CI 0.52–1.02, p=0.07) as risk factors. Similarly, a large two-sample MR study by *Vithayathil et al., 2021* including 367,561 UK Biobank participants (of which 1983 were HNC cases) found no link between BMI and HNC risk (OR = 0.98 per 1-SD higher BMI, 95% CI 0.93–1.02, p=0.35). *Larsson and Burgess, 2021* meta-analysed *Vithayathil et al., 2021* findings with results obtained using FinnGen data to increase the sample size even further (N=586,353, including 2109 cases), but still did not find a genetically predicted effect of BMI on HNC risk (OR = 0.96 per 1-SD higher BMI, 95% CI 0.77–1.19, p=0.69). With a much larger sample (N=31,523, including 12,264 cases), our IVW MR analysis suggested BMI may play a role in HNC risk, in contrast to previous studies. However, our sensitivity analyses implied that causality was uncertain.

In our study, we found some evidence that the genetically predicted effect of BMI on HNC risk was influenced by smoking. This could be due to the bidirectional relationship between smoking and adiposity reported in previous MR studies (*Taylor et al., 2019*; *Carreras-Torres et al., 2018*; *Åsvold et al., 2014*; *Freathy et al., 2011*; *Taylor et al., 2014*; *Morris et al., 2015*) or due to their shared genetic architecture (*Thorgeirsson et al., 2013*; *Wills and Hopfer, 2019*). A strength of our study was that it was the first to exploit MVMR to disentangle the effects of BMI and smoking behaviour on the risk of HNC and its subsites. An advantage of our approach compared to conducting univariable

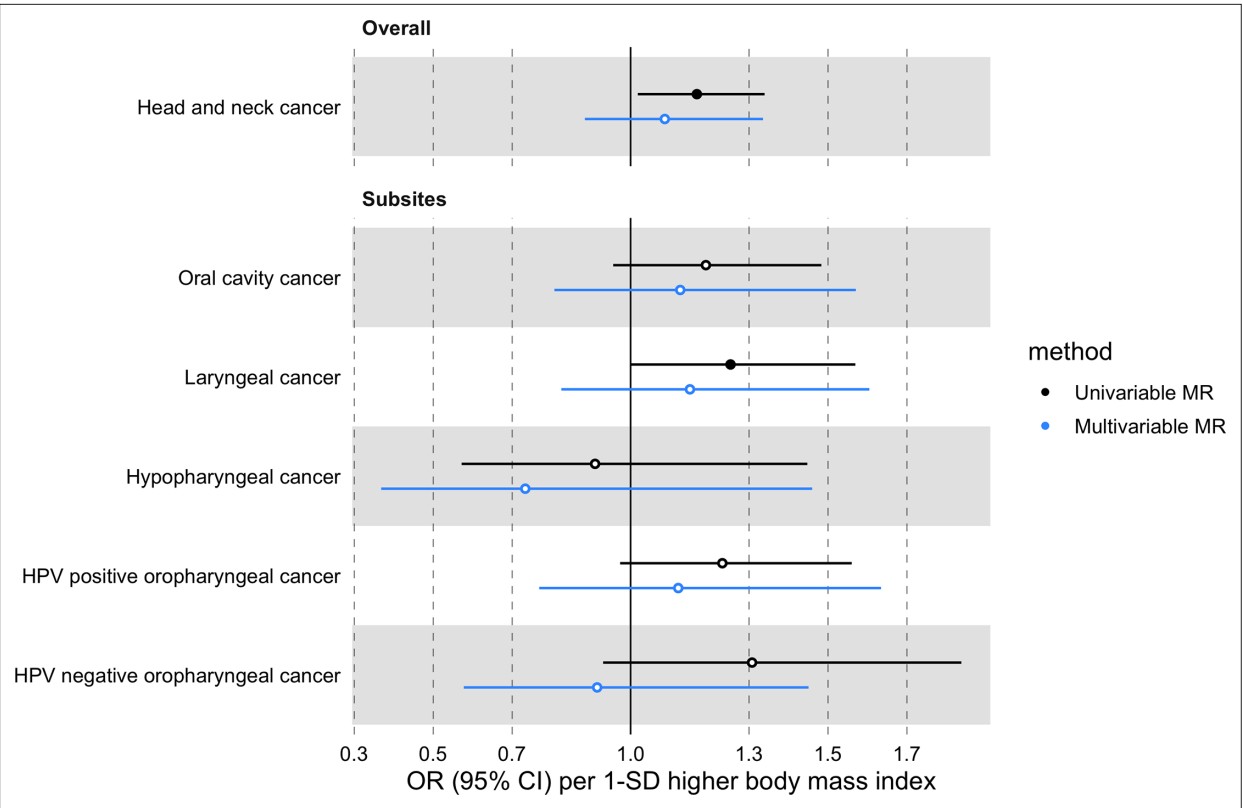

**Figure 6.** Forest plot for the genetically predicted effects of BMI on the risk of HNC and its subsites, before (univariable-black) and after (multivariable-blue) accounting for smoking initiation (SI).

MR analyses stratified by smoking status is that the former does not induce collider bias and provides estimates of direct effects irrespective of horizontal pleiotropy or mediation. Yet, we acknowledge the smoking behaviour traits used in our MVMR analyses likely capture more than just smoking, since some of the SNPs used to instrument these traits have been associated with risk-taking phenotypes and socioeconomic factors (*Gage et al., 2022*; *Schellhas et al., 2021*; *Khouja et al., 2021*; *Reed et al., 2025*). This places limits on the inferences that can be made about smoking in the context of mediation.

An important strength of our study was that the HEADSpAcE consortium GWAS used had a large sample size which conferred more statistical power to detect effects of adiposity on HNC risk compared to previous MR analyses (*Larsson and Burgess, 2021*; *Gormley et al., 2023*; *Vithayathil et al., 2021*). Furthermore, the availability of data on more HNC subsites, including oropharyngeal cancers by HPV status, allowed us to investigate the relationship between adiposity and HNC risk in more detail than previous MR studies which limited their subsite analyses to oral cavity and overall oropharyngeal cancers (*Gormley et al., 2023*; *Gui et al., 2023*). This is relevant because distinct HNC subsites are known to have different aetiologies (*Thomas et al., 2018*), although we did not find evidence of heterogeneity across subsites in our analyses investigating the genetically predicted effects of BMI, WHR, and waist circumference on HNC risk.

We acknowledge that a major limitation of MR studies, including ours, is that several untestable assumptions are required to make accurate causal inferences. It is unlikely that our findings were influenced by weak instrument bias (i.e. violating the relevance assumption) because we used strong genetic instruments to proxy our adiposity traits. However, the independence assumption of no genetic confounding and the exclusion restriction assumption of no horizontal pleiotropy could have been violated. Furthermore, we were unable to explore potential non-linear causal effects of adiposity on HNC risk in the present study.

While our study contributes valuable evidence on the role of adiposity in the development of HNC, we recognise there is a need for additional research on the subject. Our study was limited to

individuals of European ancestry, so our findings should be replicated in other ancestry groups before being generalised to non-European populations. Moreover, further research is needed to understand the biology underlying the complex relationship between smoking and adiposity, especially since it may be difficult to intervene on one without influencing the other (*Taylor et al., 2019*).

## Conclusions

In conclusion, this study indicates that adiposity does not play a major role in HNC risk. Although we did not find strong evidence of a causal effect of adiposity on HNC, obesity is an established risk factor for multiple cancers and other chronic diseases (*Larsson and Burgess, 2021*; *Lauby-Secretan et al., 2016*; *Mariosa et al., 2019*). Hence, there is still value in aiming to reduce the levels of excess adiposity in the population.

## Acknowledgements

We thank Richard Wilkinson for proofreading several versions of the manuscript. We would also like to thank Weili (Jason) Qiu, the IEU Data Manager, for his help debugging code and uploading the HNC GWAS summary statistics to the IEU OpenGWAS platform. FMB was supported by a Wellcome Trust PhD studentship in Molecular, Genetic and Lifecourse Epidemiology (224982/Z/22/Z). RCR was supported by a Cancer Research UK grant (C18281/A29019). MCB is supported by a University of Bristol Vice Chancellor's Fellowship, the British Heart Foundation (AA/18/1/34219) and the UK Medical Research Council (MC_UU_00032/5). GDS works within the MRC Integrative Epidemiology Unit at the University of Bristol, which is supported by the Medical Research Council (MC_UU_00032/1). CLR was supported by the Medical Research Council (MC_UU_00011/5) and by a Cancer Research UK (C18281/A29019) programme grant (the Integrative Cancer Epidemiology Programme). SV was funded by an EU Horizon 2020 grant (agreement number 825771) and NIDCR National Institutes of Dental and Craniofacial Health (R03DE030257). JK works in a unit that receives support from the University of Bristol, a Cancer Research UK grant (C18281/A29019) and the UK Medical Research Council (grant number: MC_UU_00032/7).

## Additional information

### Competing interests

Tom G Richardson: TGR is an employee of GlaxoSmithKline outside of this work. The other authors declare that no competing interests exist.

### Funding

| Funder | Grant reference number | Author |
| --- | --- | --- |
| Wellcome Trust | 10.35802/224982 | Fernanda Morales Berstein |
| Cancer Research UK | C18281/A29019 | Rebecca C Richmond Jasmine Khouja Caroline L Relton |
| University of Bristol Vice Chancellor's Fellowship | | M Carolina Borges |
| British Heart Foundation | AA/18/1/34219 | M Carolina Borges |
| Medical Research Council | MC_UU_00032/1 | George Davey Smith |
| Medical Research Council | MC_UU_00011/5 | Caroline L Relton |
| EU Horizon 2020 grant | Agreement number 825771 | Shama Virani |
| NIDCR National Institutes of Dental and Craniofacial Health | R03DE030257 | Shama Virani |
| Medical Research Council | MC_UU_00032/7 | Jasmine Khouja |

| Funder | Grant reference number | Author |
| --- | --- | --- |
| Medical Research Council | MC_UU_00032/5 | M Carolina Borges |

The funders had no role in study design, data collection and interpretation, or the decision to submit the work for publication. For the purpose of Open Access, the authors have applied a CC BY public copyright license to any Author Accepted Manuscript version arising from this submission.

## Author contributions

Fernanda Morales Berstein, Conceptualization, Formal analysis, Visualization, Writing - original draft, Writing – review and editing; Jasmine Khouja, Mark Gormley, Conceptualization, Writing – review and editing; Elmira Ebrahimi, Shama Virani, James D McKay, Paul Brennan, Tom G Richardson, Caroline L Relton, George Davey Smith, Writing – review and editing; M Carolina Borges, Supervision, Writing – review and editing; Tom Dudding, Rebecca C Richmond, Conceptualization, Supervision, Writing – review and editing

## Author ORCIDs

Fernanda Morales Berstein ● https://orcid.org/0000-0002-8237-2021
Jasmine Khouja ● http://orcid.org/0000-0002-7944-2981
Mark Gormley ● https://orcid.org/0000-0001-5733-6304
Elmira Ebrahimi ● https://orcid.org/0000-0003-0910-3447
Shama Virani ● http://orcid.org/0000-0002-1163-432X
Tom G Richardson ● https://orcid.org/0000-0002-7918-2040
Caroline L Relton ● https://orcid.org/0000-0003-2052-4840
George Davey Smith ● https://orcid.org/0000-0002-1407-8314
M Carolina Borges ● http://orcid.org/0000-0001-7785-4547
Tom Dudding ● https://orcid.org/0000-0003-3756-040X
Rebecca C Richmond ● http://orcid.org/0000-0003-0574-5071

## Ethics

Human subjects: This study is a secondary data analysis using publicly available datasets. Therefore, it did not require ethical approval. Informed consent and ethical approval were obtained by the authors of the genome-wide association studies (GWASs) included in our analyses. Details can be found in each GWAS publication.

Reviewer #1 (Public review): https://doi.org/10.7554/eLife.106075.3.sa1
Author response https://doi.org/10.7554/eLife.106075.3.sa2

# Additional files

## Supplementary files

Supplementary file 1. Supplementary Tables A to N.

MDAR checklist

Reporting standard 1. STROBE-MR checklist.

## Data availability

All the GWAS datasets used in our study are publicly available. The GWAS summary statistics for waist circumference are available via the IEU OpenGWAS platform (id: ukb-b-9405). The GWAS summary statistics for BMI and WHR by *Pulit et al., 2019* can be downloaded from https://zenodo.org/records/1251813. The data sources for the other adiposity-related measures have been specified in Table 1. The smoking behaviour traits GWAS data were downloaded from https://data.bris.ac.uk/data/dataset/10i96zb8gm0j81yz0q6ztei23d (for CSI) and https://doi.org/10.13020/przg-dp88 (for smoking initiation). The outcome datasets used in our analyses have been uploaded to the IEU OpenGWAS project platform for reproducibility. However, because the data was originally in build GRCh38, some multiallelic SNPs that could not be aligned with GRCh37 Human Genome reference sequence were dropped when lifting the data to build HG19/GRCh37 (which was required at the time of upload: April 2024). The following IEU OpenGWAS id's were assigned to the European HEADSpAcE HNC GWAS

datasets including/excluding UK Biobank: ieu-b-5129/ieu-b-5123 for overall HNC, ieu-b-5132/ieu-b-5126 for oral cavity cancer, ieu-b-5130/ieu-b-5124 for hypopharynx cancer, ieu-b-5134/ieu-b-5128 for HPV positive oropharynx cancer, ieu-b-5133/ieu-b-5127 for HPV negative oropharynx cancer, and ieu-b-5131/ieu-b-5125 for larynx cancer. The R code used to run the MR analyses is available at https://github.com/fernandam93/adiposity_HNC_MR (copy archived at *Morales Berstein, 2025*).

The following previously published datasets were used:

| Author(s) | Year | Dataset title | Dataset URL | Database and Identifier |
|---|---|---|---|---|
| Ried et al. | 2016 | Summary association statistics for body shape PC1 | https://504394d8-624a-4827-9f25-95a83cd9675a.filesusr.com/archives/1d0101_b69b38ae716f45e9b260500e74f4969a.gz?dn=GIANT_metal_result_bodyshape_pc1_all_iv_hetero_20111006_adjusted1.txt.gz | Hirschhorn Lab - GIANT Consortium results, GIANT_metal_result_bodyshape_pc1_all_iv_hetero_20111006_adjusted1.txt.gz |
| Ried et al. | 2016 | Summary association statistics for body shape PC2 | https://504394d8-624a-4827-9f25-95a83cd9675a.filesusr.com/archives/1d0101_37b0e22162bb410a9ddaaada7343a8db.gz?dn=GIANT_metal_result_bodyshape_pc2_all_iv_hetero_20111006_adjusted1.txt.gz | Hirschhorn Lab - GIANT Consortium results, GIANT_metal_result_bodyshape_pc2_all_iv_hetero_20111006_adjusted1.txt.gz |
| Ried et al. | 2016 | Summary association statistics for body shape PC3 | https://504394d8-624a-4827-9f25-95a83cd9675a.filesusr.com/archives/1d0101_846cc20a42c14816ab262c763142de25.gz?dn=GIANT_metal_result_bodyshape_pc3_all_iv_hetero_20111006_adjusted1.txt.gz | Hirschhorn Lab - GIANT Consortium results, GIANT_metal_result_bodyshape_pc3_all_iv_hetero_20111006_adjusted1.txt.gz |
| Ried et al. | 2016 | Summary association statistics for body shape PC4 | https://504394d8-624a-4827-9f25-95a83cd9675a.filesusr.com/archives/1d0101_8f21ef241c454e0c816d34e886a82719.gz?dn=GIANT_metal_result_bodyshape_pc4_all_iv_hetero_20111006_adjusted1.txt.gz | Hirschhorn Lab - GIANT Consortium results, GIANT_metal_result_bodyshape_pc4_all_iv_hetero_20111006_adjusted1.txt.gz |
| Pulit SL | 2018 | Summary-level data from meta-analysis of fat distribution phenotypes in UK Biobank and GIANT | https://doi.org/10.5281/zenodo.1251813 | Zenodo, 10.5281/zenodo.1251813 |
| Elsworth B | 2018 | Waist circumference | https://opengwas.io/datasets/ukb-b-9405 | IEU OpenGWAS, ukb-b-9405 |

*Continued on next page*

*Continued*

| Author(s) | Year | Dataset title | Dataset URL | Database and Identifier |
|---|---|---|---|---|
| Elsworth B | 2018 | Body fat percentage | https://opengwas.io/datasets/ukb-b-8909 | IEU OpenGWAS, ukb-b-8909 |
| Wootton et al. | 2019 | Genome-wide association study of lifetime smoking index in a sample of 462,690 individuals from the UK Biobank | https://doi.org/10.5523/bris.10i96zb8gm0j81yz0q6ztei23d | University of Bristol Life Sciences, 10.5523/bris.10i96zb8gm0j81yz0q6ztei23d |
| Richardson T | 2020 | Comparative body size at age 10, Males and Females | https://opengwas.io/datasets/ieu-b-5107 | IEU OpenGWAS, ieu-b-5107 |
| Martin et al. | 2021 | Genetic Evidence for Different Adiposity Phenotypes and their Opposing Influence on Ectopic Fat and Risk of Cardiometabolic Disease | https://doi.org/10.2337/figshare.14555463.v1 | American Diabetes Association, 10.2337/figshare.14555463.v1 |
| Saunders et al. | 2022 | Data related to Genetic diversity fuels gene discovery for tobacco and alcohol use | https://conservancy.umn.edu/items/91f6a003-6af2-4809-9785-53dc579dc788 | Data Repository for the University of Minnesota, 10.13020/przg-dp88 |
| Dudding T | 2025 | Head and neck cancer (excl. UKB) | https://opengwas.io/datasets/ieu-b-5123 | IEU OpenGWAS, ieu-b-5123 |
| Dudding T | 2025 | Hypopharyngeal cancer (excl. UKB) | https://opengwas.io/datasets/ieu-b-5124 | IEU OpenGWAS, ieu-b-5124 |
| Dudding T | 2025 | Laryngeal cancer (excl. UKB) | https://opengwas.io/datasets/ieu-b-5125 | IEU OpenGWAS, ieu-b-5125 |
| Dudding T | 2025 | Oral cavity cancer (excl. UKB) | https://opengwas.io/datasets/ieu-b-5126 | IEU OpenGWAS, ieu-b-5126 |
| Richardson T | 2020 | Adult BMI sex-combined | https://opengwas.io/datasets/ieu-b-5118 | IEU OpenGWAS, ieu-b-5118 |
| Dudding T | 2025 | HPV negative oropharyngeal cancer (excl. UKB) | https://opengwas.io/datasets/ieu-b-5127 | IEU OpenGWAS, ieu-b-5127 |
| Dudding T | 2025 | HPV positive oropharyngeal cancer (excl. UKB) | https://opengwas.io/datasets/ieu-b-5128 | IEU OpenGWAS, ieu-b-5128 |

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

# Appendix 1

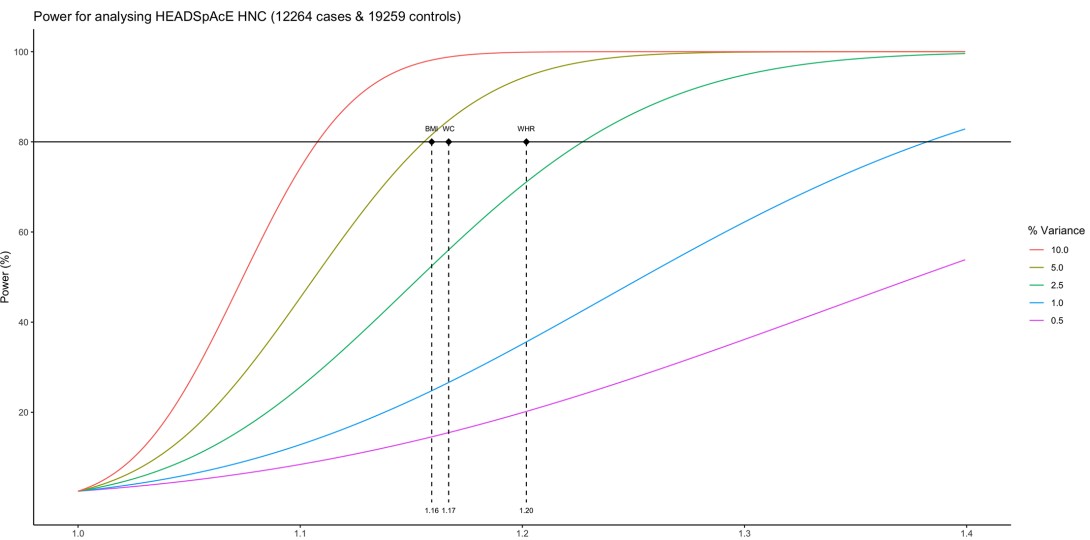

**Appendix 1—figure 1.** Power calculations for HEADSpAcE head and neck cancer risk. Minimum odds ratios for body mass index (BMI), waist circumference (WC), and waist-to-hip ratio (WHR) were estimated assuming an alpha value of 0.05, power of 80% and total variances ($R^2$) of 4.8%, 3.1%, and 4.4%, respectively.

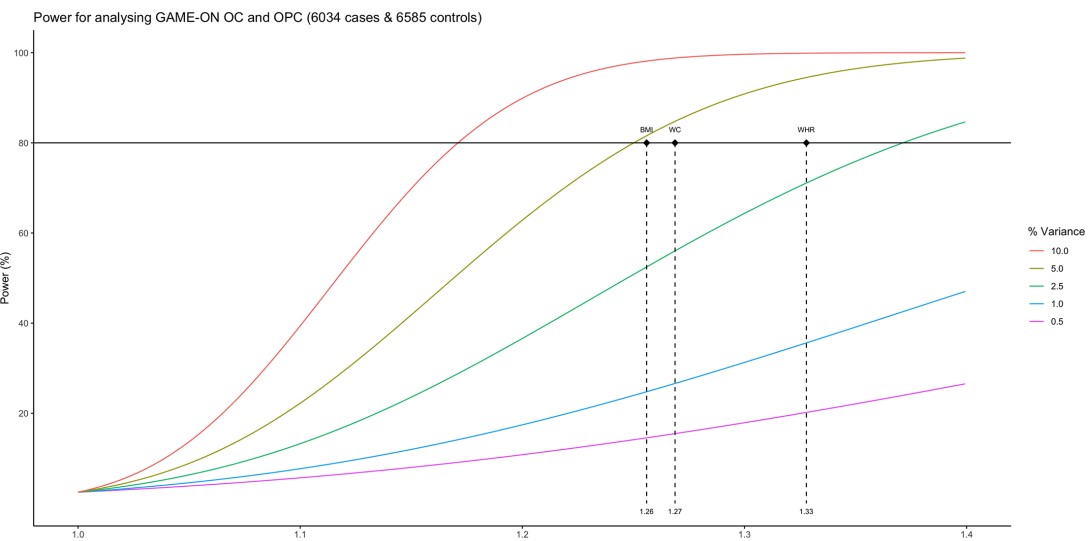

**Appendix 1—figure 2.** Power calculation plot for GAME-ON oral cancer and oropharyngeal cancer risk. Minimum odds ratios for body mass index (BMI), waist circumference (WC), and waist-to-hip ratio (WHR) were estimated assuming an alpha value of 0.05, power of 80% and total variances ($R^2$) of 4.8%, 3.1%, and 4.4%, respectively.

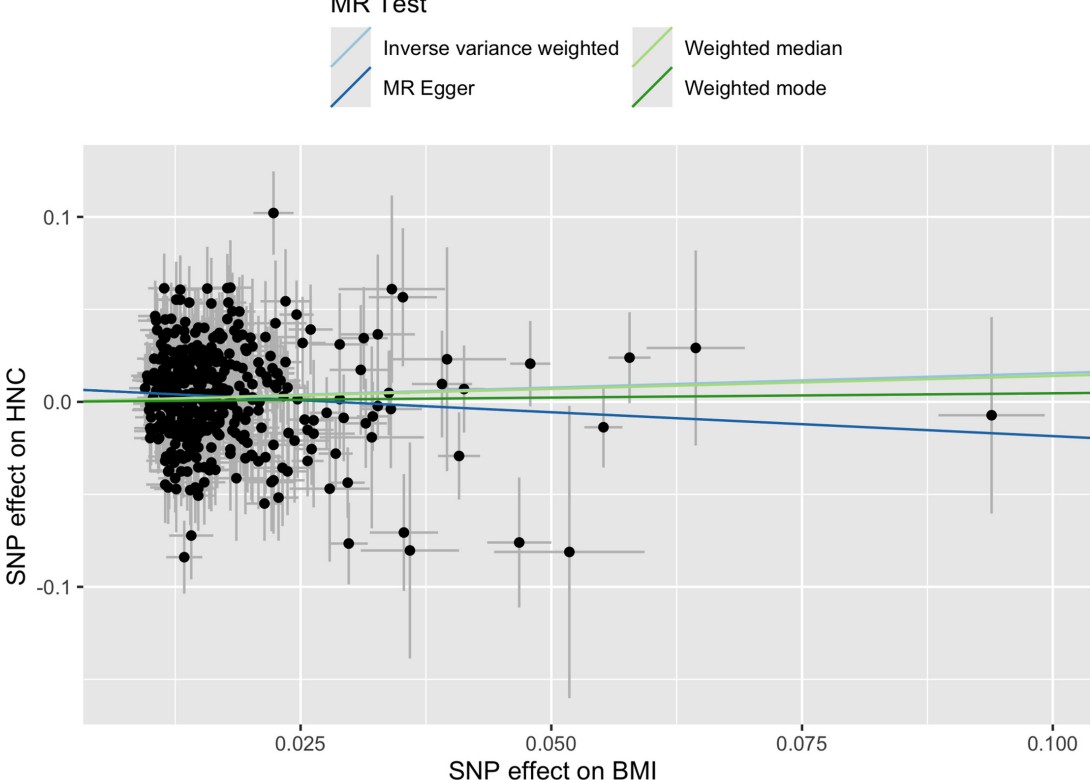

**Appendix 1—figure 3.** Scatter plot for the genetically predicted effects of body mass index (BMI) on the risk of head and neck cancer (HNC).

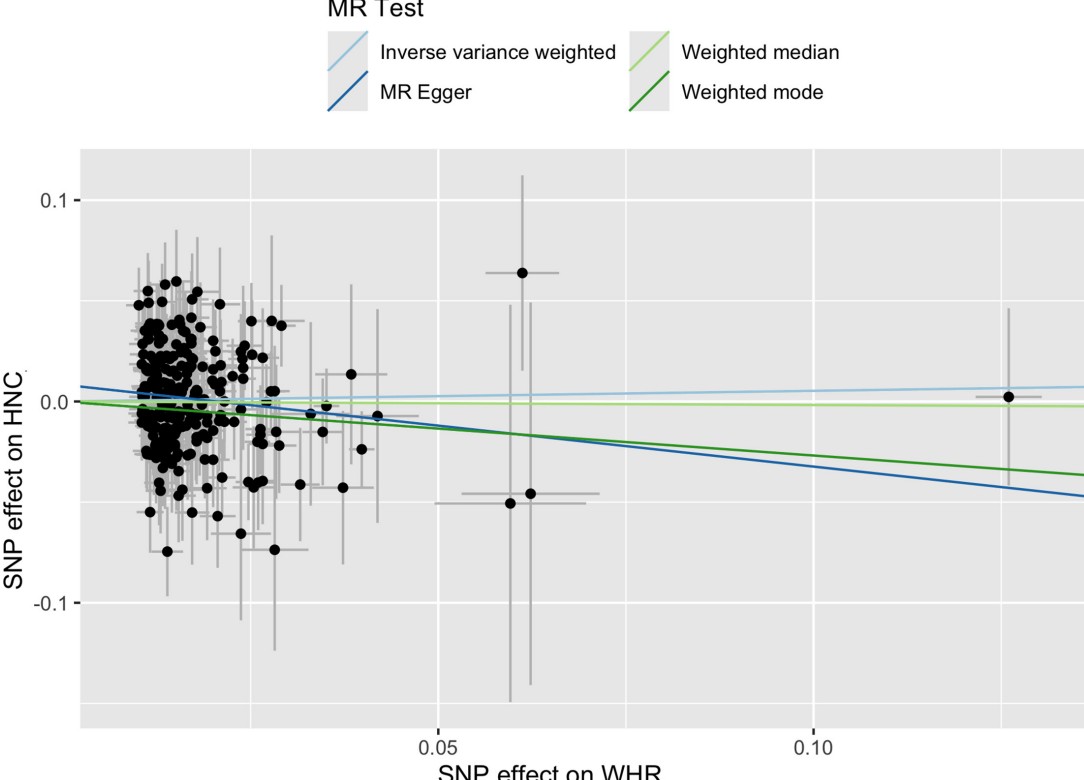

**Appendix 1—figure 4.** Scatter plot for the genetically predicted effects of waist-to-hip ratio (WHR) on the risk of head and neck cancer (HNC).

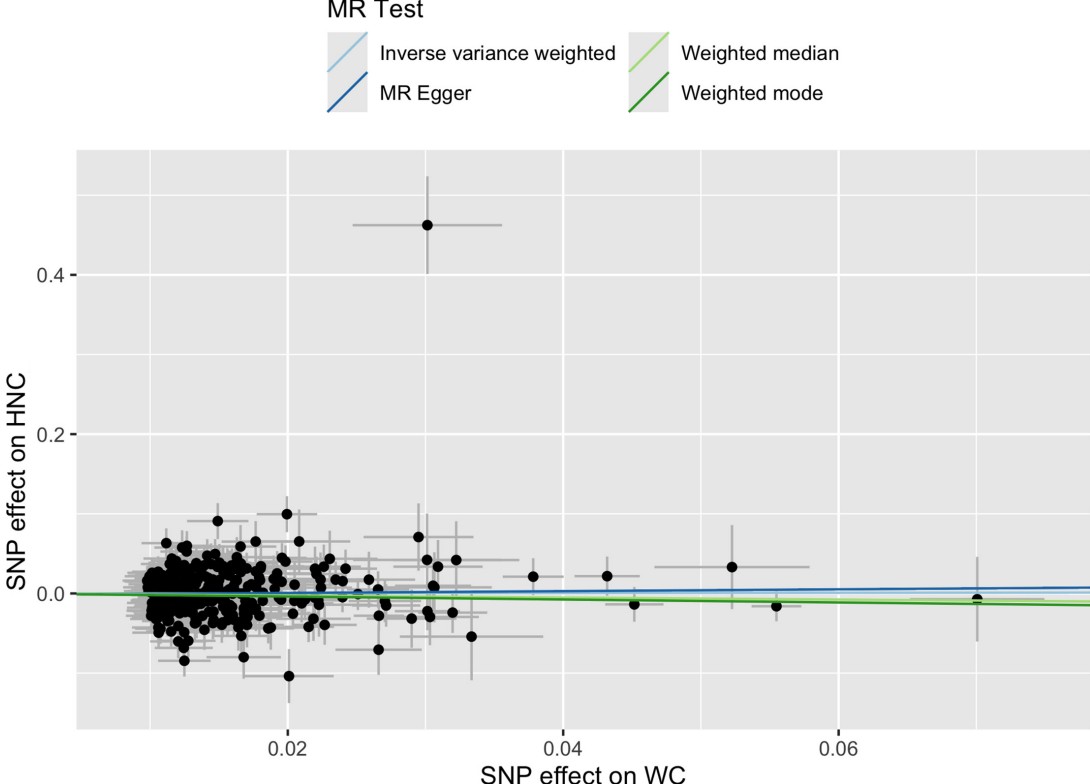

**Appendix 1—figure 5.** Scatter plot for the genetically predicted effects of waist circumference (WC) on the risk of head and neck cancer (HNC).

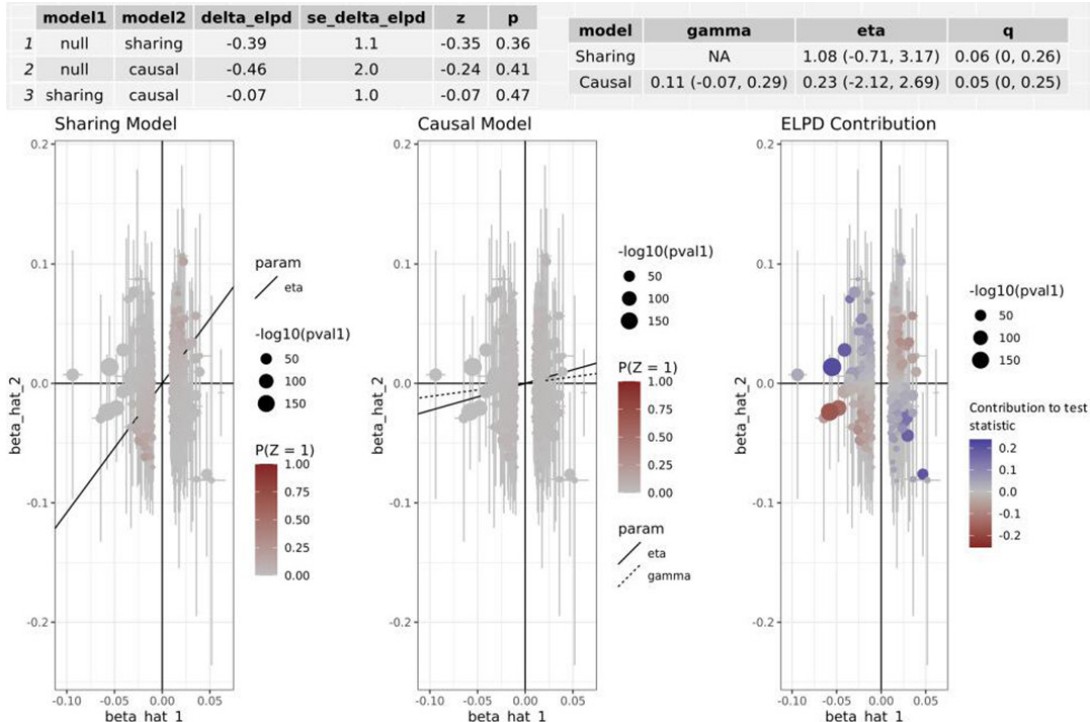

| | model1 | model2 | delta_elpd | se_delta_elpd | z | p |
|---|---|---|---|---|---|---|
| 1 | null | sharing | -0.39 | 1.1 | -0.35 | 0.36 |
| 2 | null | causal | -0.46 | 2.0 | -0.24 | 0.41 |
| 3 | sharing | causal | -0.07 | 1.0 | -0.07 | 0.47 |

| model | gamma | eta | q |
|---|---|---|---|
| Sharing | NA | 1.08 (-0.71, 3.17) | 0.06 (0, 0.26) |
| Causal | 0.11 (-0.07, 0.29) | 0.23 (-2.12, 2.69) | 0.05 (0, 0.25) |

**Appendix 1—figure 6.** CAUSE analysis for the genetically predicted effect of body mass index (BMI) on head and neck cancer (HNC) risk. CAUSE estimates for HNC reported per 1-SD higher BMI in log odds ratio scale. The ELPD Contribution plot shows the relative contribution of each SNP to the CAUSE test statistic. Only SNPs with p<5e-8 are shown. SNPs represented by larger circles reflect smaller p-values for the associations between genetic variants and BMI. SNPs that contribute more to the causal model are shown in warmer tones (i.e. red), while those that contribute more to the sharing model are shown in colder tones (i.e. blue). The delta_elpd is the statistic used to compare models. It is equal to elpd(model 1)- elpd(model 2). In the table on the left, negative delta_elpd's suggest that model 2 is a better fit to the data than model 1 (i.e. that the sharing model is better than the null model in row 1, that the causal model is better than the null model in row 2, and that the causal model is better than the sharing model in row 3). The corresponding p-values test whether model 2 is a better fit than model 1. Here, row 3 suggests that the causal model is not a much better fit than the sharing model (the delta_elpd is negative but the p-value is 0.47, so there is no overwhelming evidence against the null hypothesis that the causal model is better than the sharing model). In the table on the right, eta represents the sharing factor effect (SNPs affect shared factor and shared factor simultaneously affects BMI and HNC) and gamma represents the causal factor effect (SNPs affect BMI and BMI affects HNC). Here, '0.11 (-0.07, 0.29)' represents the genetically predicted effect of BMI on HNC after adjusting for correlated and uncorrelated horizontal pleiotropy (results in log odds ratio scale). The intervals shown are credible intervals.

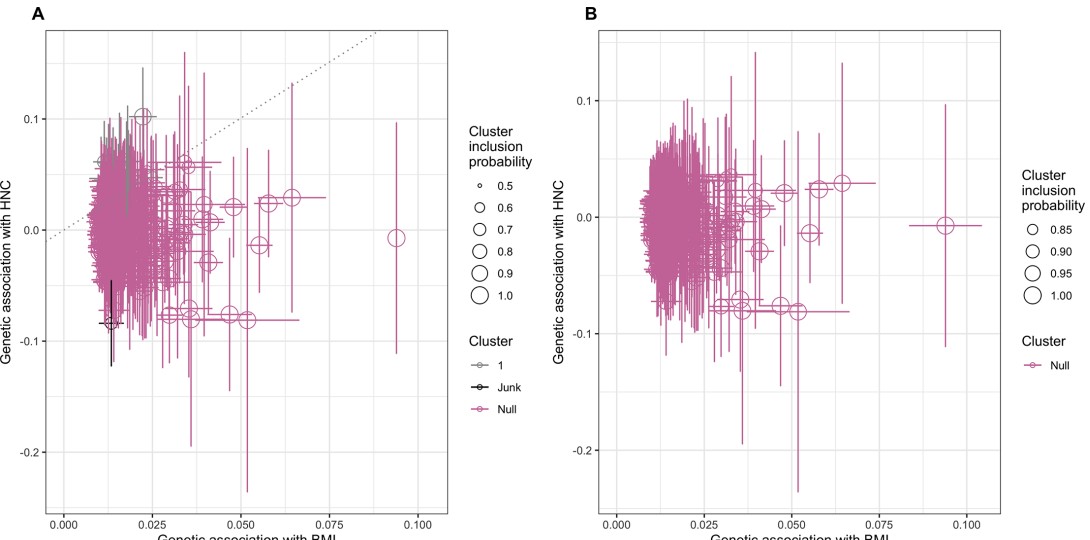

**Appendix 1—figure 7.** Scatter plots depicting clusters of genetic associations with body mass index (BMI) and head and neck cancer (HNC), before (**A**) and after (**B**) conditional probability filtering.

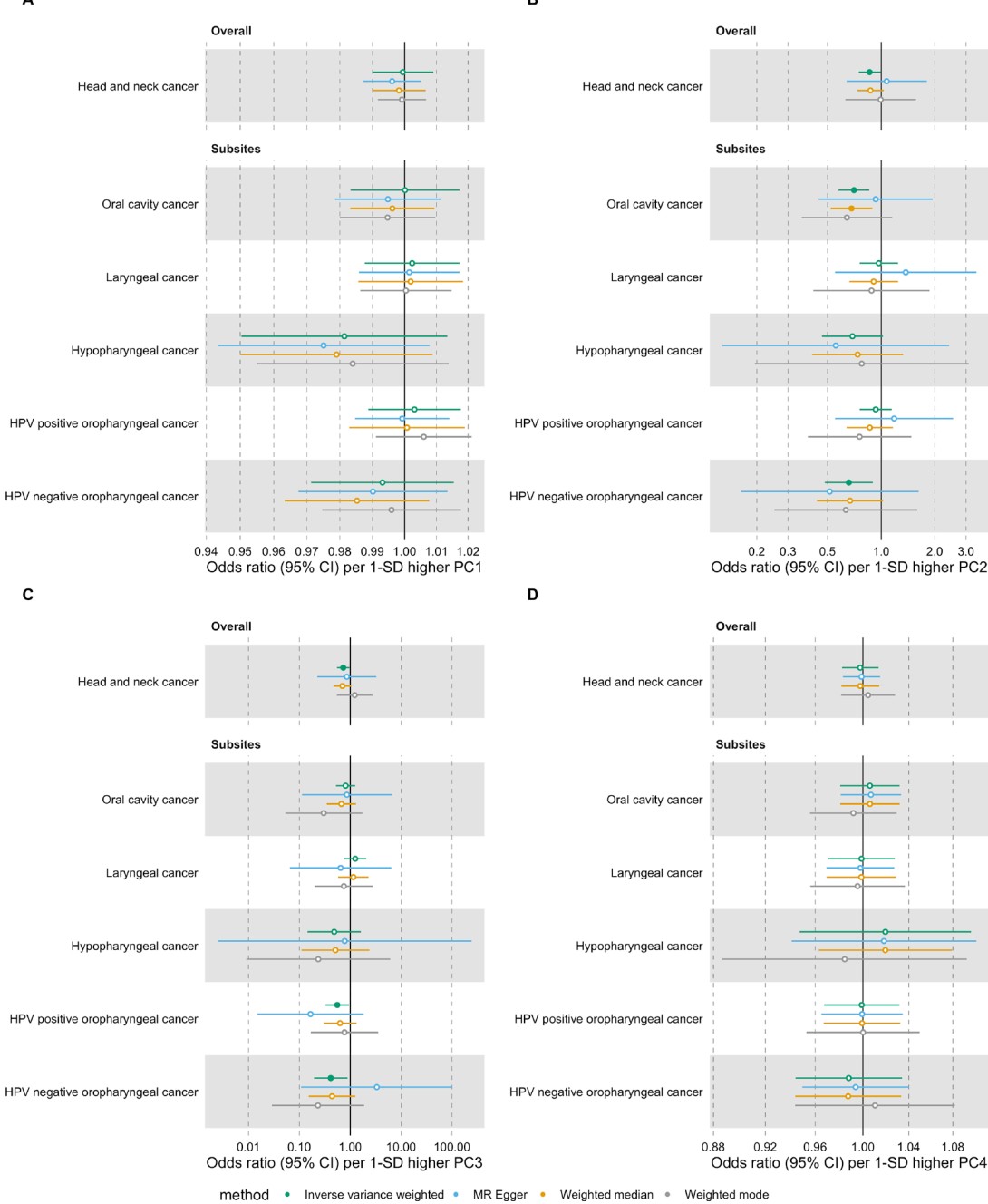

**Appendix 1—figure 8.** Forest plots for the genetically predicted effects of four body shape principal components (PCs) on the risk of head and neck cancer and its subsites, where (**A**) PC1 is a measure of overall adiposity, (**B**) PC2 is a measure of tall and slim vs short and plump, (**C**) PC3 is a measure of tall with small hip vs short with big hip and (**D**) PC4 is a measure of high body mass index (BMI) and weight with small hip and waist vs low BMI and weight with big hip and waist.

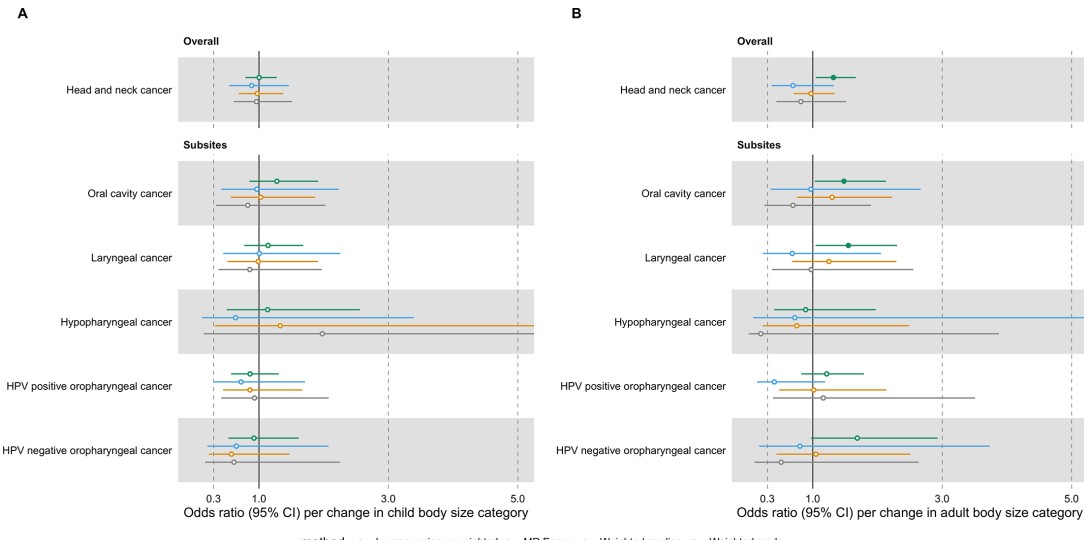

**Appendix 1—figure 9.** Forest plots for the genetically predicted effects of (**A**) childhood and (**B**) adulthood body size on the risk of head and neck cancer and its subsites.

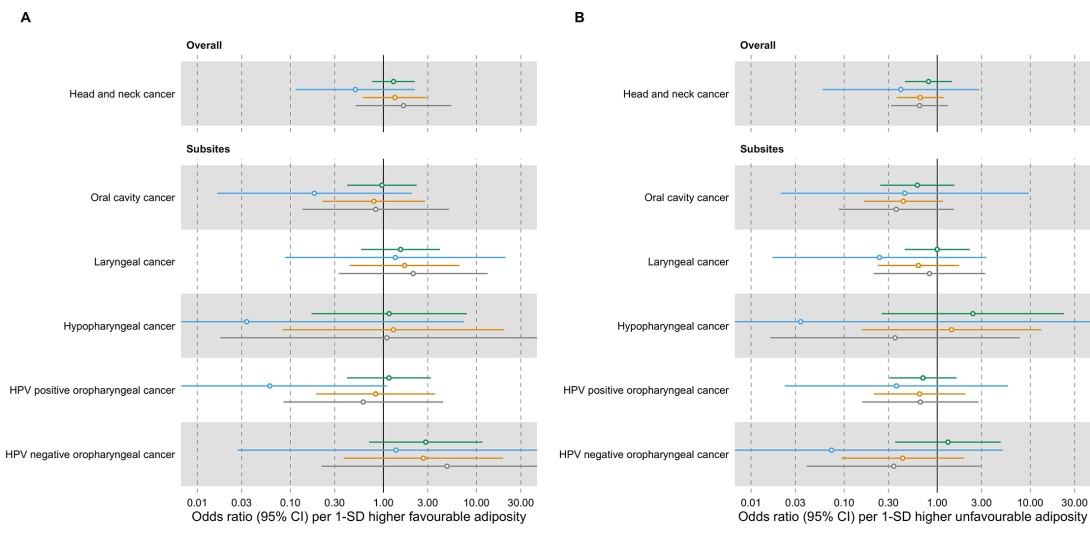

**Appendix 1—figure 10.** Forest plots for the genetically predicted effects of (**A**) favourable and (**B**) unfavourable adiposity on the risk of head and neck cancer and its subsites.

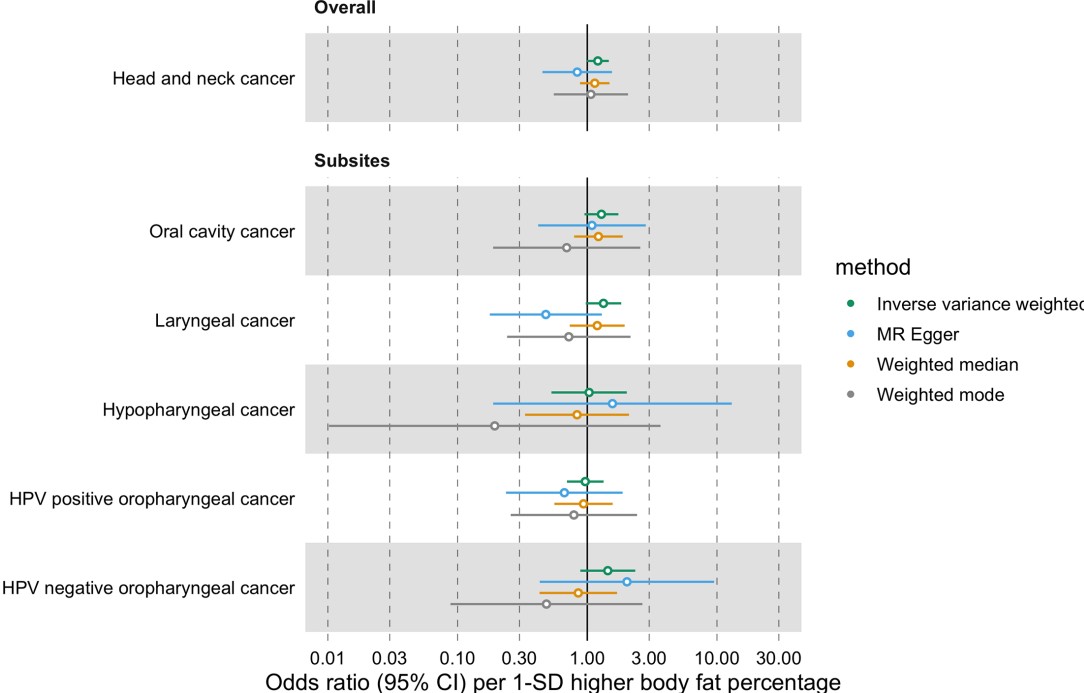

**Appendix 1—figure 11.** Forest plots for the genetically predicted effect of body fat percentage on the risk of head and neck cancer and its subsites.

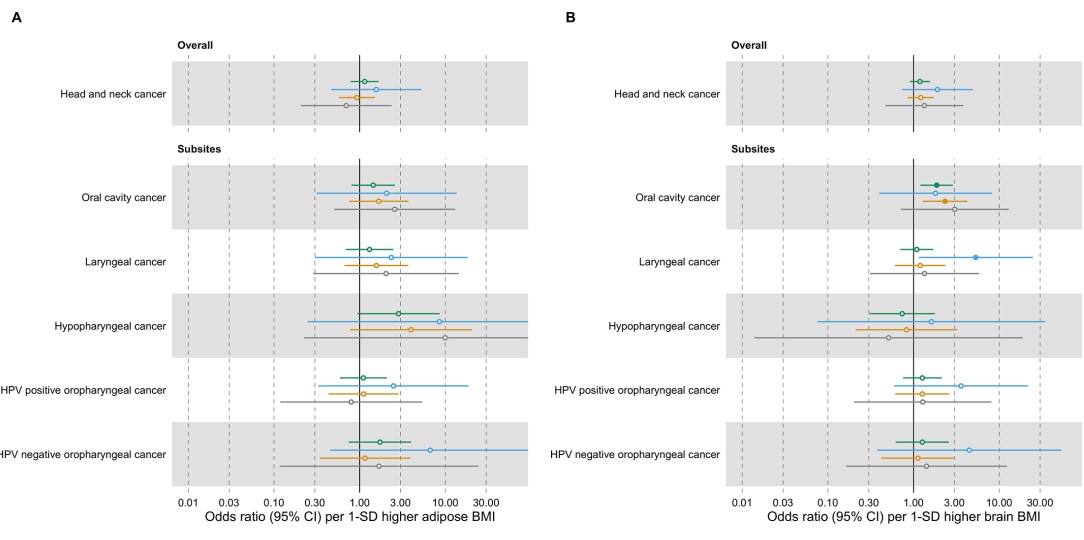

**Appendix 1—figure 12.** Forest plots for genetically predicted effects of (**A**) adipose and (**B**) brain tissue-specific body mass index (BMI) on head and neck cancer and its subsites.

