## [Editor Report · eLife Assessment]

This **important** study reports that higher genetically predicted adiposity is not strongly associated with the risk of head and neck cancer. The **convincing** evidence is supported by rigorous Mendelian Randomization approaches, using multiple genetic instruments and models that reduce sensitivity to pleiotropy. The work will be of interest to researchers studying cancer risk factors and genetic epidemiology.

---

## [Referee Report · Reviewer #1 (Public review)]

Summary:

The authors have conducted the largest to date Mendelian Randomization (MR) analysis of the association between genetically predicted measures of adiposity and risk of head and neck cancer (HNC) overall and by subsites within HNC. MR uses genetic predictors of an exposure, such as gene variants associated with high BMI or tobacco use, rather than data from individual physical exams or questionnaires and if it can be done in its idealized state, there should be no problems with confounding. Traditional epidemiologic studies have reported a variety of associations between BMI (and a few other measures of adiposity) and risk of HNC that typically differs by the smoking status of the subjects. Those findings are controversial given the complex relationship between tobacco and both BMI and HNC risk. Tobacco smokers are often thinner than no-smokers so this could create an artificial ('confounded') association that may not be fully adjusted away in risk models. The findings of a BMI-HNC association are often attributed to residual confounding and this seems ripe for an MR approach if suitable genetic instrumental variables can be created. Here the authors built a variety of genetic instrumental variables for BMI and other measures of adiposity as well as two instrumental variables for smoking habits and then tested their hypotheses in a large case-controls set of HNC and controls with genetic data.

The authors found that the genetic model for BMI was associated with HNC risk in simple models, but this association disappeared when using models that better accounted for pleiotropy, the condition when genetic variants are associated with more than one trait such as both BMI and tobacco use. When they used both adiposity and tobacco use genetic instruments in a single model, there was a strong association with genetically predicted tobacco use (as is expected) but there was no remaining association with genetic predictors of adiposity. They conclude that high BMI/adiposity is not a risk factor for HNC.

Strengths:

The primary strength was the expansive use of a variety of different genetic instruments for BMI/adiposity/body size along with employing a variety of MR model types, several of which are known to be less sensitive to pleiotropy. They also used the largest case-control sample size to date.

Weaknesses:

The lack of pleiotropy is an unconfirmable assumption of MR and the addition of those models is therefore quite important as this is a primary weakness of the MR approach. Given that concern, I read the sensitivity analyses using pleiotropy-robust models as the main result and in that case, they are more limited in their ability to test their hypothesis as these models do not show a robust BMI instrumental variable association.

Comments on the revised manuscript:

After the first round of review, the authors have improved the manuscript by (1) adding the requested power calculations and adding text to help the reader integrate that additional information; (2) adding the main effects for the tobacco instruments; (3) updating the comparison of their results to the prior literature; (4) and some other edits to the text. They have declined to include the smoking stratified estimates and provide a rationale for this decision that references the potential for collider bias. While true that yet another bias might be introduced, that gets added to the list and the careful reader would know that. Many important questions in cancer etiology can only be addressed via observational approaches and each observational approach has the potential for a long list of biases. The best inference comes from integrating the totality of the data and realizing that most conclusions are subject to updating as we conduct more work and learn more.

---

## [Author Response]

The following is the authors’ response to the original reviews.

**Joint Public Review:**
Weaknesses:The lack of pleiotropy is an unconfirmable assumption of MR, and the addition of those models is therefore quite important, as this is a primary weakness of the MR approach. Given that concern, I read the sensitivity analyses using pleiotropy-robust models as the main result, and in that case, they can't test their hypotheses as these models do not show a BMI instrumental variable association. The other weakness, which might be remedied, is that the power of the tests here is not described. When a hypothesis is tested with an under-powered model, the apparent lack of association could be due to inadequate sample size rather than a true null. Typically, when a statistically significant association is reported, power concerns are discounted as long as the study is not so small as to create spurious findings. That is the case with their primary BMI instrumental variable model - they find an association so we can presume it was adequately powered. But the primary models they share are not the pleiotropy-robust methods MR-Egger, weighted median, and weighted mode. The tests for these models are null, and that could mean a couple of things: (1) the original primary significant association between the BMI genetic instrument was due to pleiotropy, and they therefore don't have a robust model to explore the effects of the tobacco genetic instrument. (2) The power for the sensitivity analysis models (the pleiotropy-robust methods) is inadequate, and the authors share no discussion about the relative power of the different MR approaches. If they do have adequate power, then again, there is no need to explore the tobacco instrument.Reviewing Editor Comments:We suggest that the authors add power estimates to assess whether the sample size is sufficient, given the strength and variability of the genetic instruments. It would also be helpful to present effect estimates for the tobacco instruments alone, to clarify their independent contribution and improve the interpretation of the joint models. In addition, the role of pleiotropy should be addressed more clearly, including which model is considered primary. Stratified analyses by smoking status are encouraged, as prior studies indicate that BMI-HNC associations may differ between smokers and non-smokers. Finally, the comparison with previous studies should be revised, as most reported null findings without accounting for tobacco instruments. If this study finds an association, it should not be framed as a replication

We would like to highlight that post-hoc power calculations are often considered redundant since the statistical power estimated for an observed association is directly related to its p-value[1]. In other words, the uncertainty of the association is already reflected in its 95% confidence interval. However, we understand power calculations may still be of interest to the reader, so we have incorporated them in the revised manuscript. We have edited the text as follows (lines 151-155):“Consequently, we used the total R^2^ values to examine the statistical power in our study[42]. However, we acknowledge that the value of post-hoc power calculations is limited, since the statistical power estimated for an observed association is already reflected in the 95% confidence interval presented alongside the point estimate[43].” We have also added supplementary figures 1 and 2.

We can see that when using the latest HEADSpAcE data we were able to detect BMI-HNC ORs as small as 1.16 with 80% power, while the GAME-ON dataset only permitted the detection of ORs as small as 1.26 using the same BMI instruments (Figure B). We have explained these figures in the results section as follows (lines 257-263): “Using the BMI genetic instruments (total R^2^ = 4.8%) and an α of 0.05, we had 80% statistical power to detect an OR as small as 1.16 for HNC risk (Supplementary Figure 1). For WHR (total R^2^ = 3.1%) and WC (total R^2^ = 4.4%), we could detect odds ratios (ORs) as small as 1.20 and 1.17, respectively. This is an improvement in terms of statistical power compared to the GAME-ON analysis published by Gormley et al.[28], for which there was 80% power to detect an OR as small as 1.26 using the same BMI genetic instruments (Supplementary Figure 2).”

The reason we use inverse variance weighted (IVW) Mendelian randomization (MR) to obtain our main results rather than the pleiotropy-robust methods mentioned by the reviewer/editors (i.e., MR-Egger, weighted median and weighted mode) is that the former has greater statistical power than the latter[2]. Hence, instead of focussing on the statistical significance of the pleiotropy-robust analyses, we consider it is of more value to compare the consistency of the effect sizes and direction of the effect estimates across methods. Any evidence of such consistency increases our confidence in our main findings, since each method relies on different assumptions. As we cannot be sure about the presence and nature of horizontal pleiotropy, it is useful to compare results across methods even though they are not equally powered. It is true that our results for the genetically predicted effects of body mass index (BMI) on the risk of head and neck cancer (HNC) differ across methods. This is precisely what led us to question the validity of our main finding (suggesting a positive effect of BMI on HNC risk). We have now clarified this in the methods section of the revised manuscript as advised. Lines 165-171:

“Because the IVW method assumes all genetic variants are valid instruments[44], which is unlikely the case, three pleiotropy-robust two-sample MR methods (i.e., MR-Egger[45], weighted median[46] and weighted mode[47]) were used in sensitivity analyses. When the magnitude and direction of effect estimates are consistent across methods that rely on different assumptions, the main findings are more convincing. As we cannot be sure about the presence and nature of horizontal pleiotropy, it is useful to compare results across methods even if they are not equally powered.”

We understand that the reviewer/editors are concerned that we do not have a robust model to explore the role of tobacco consumption in the link between BMI and HNC. However, we have a different perspective on the matter. If indeed, the main IVW finding for BMI and HNC is due to pleiotropy (since some of the pleiotropy-robust methods suggest conflicting results), then the IVW multivariable MR method is a way to explore the potential source of this bias[3]. We were particularly interested in exploring the role of smoking in the observed association because smoking and adiposity are known to influence each other [4-9] and share a genetic basis[10, 11].

We agree that it would be useful to present the univariable MR effect estimates for smoking behaviour and HNC risk along those obtained using multivariable MR. We have now included the univariable MR estimates for both smoking behaviour variables as a note under Supplementary Table 11 and in the manuscript (lines 316-318): “In univariable IVW MR, both CSI and SI were linked to an increased risk of HNC (CSI OR=4.47 per 1-SD higher CSI, 95%CI 3.31–6.03, p<0.001; SI OR=2.07 per 1-SD higher SI 95%CI 1.60–2.68, p<0.001) (Additional File 2: note in Supplementary Table 11).”

We understand the appeal of conducting stratified MR analyses by smoking status. However, we anticipate such analyses would hinder the interpretation of our findings as they can induce collider bias which could spuriously lead to different effect estimates across strata[12, 13].

We thank the reviewer/editors for their comment regarding the way we frame of our findings. We have now edited the discussion section to highlight our study results are different to those obtained in studies that do not account for smoking behaviour. Lines 398-401: “With a much larger sample (N=31,523, including 12,264 cases), our IVW MR analysis suggested BMI may play a role in HNC risk, in contrast to previous studies. However, our sensitivity analyses implied that causality was uncertain.”

**Reviewer #1 (Recommendations for the authors):**
The authors do share a table of the percent variance explained of the different genetic instruments, which vary widely, and that table is very welcome because we can get some sense of their utility. The problem is that they don't translate that into a power estimate for the case-control study size that they use. They say that it is the biggest to date, which is good, but without some formal power estimate, it is not particularly reassuring. A framework for MR study power estimates was reported in PMID: 19174578, but that was using very simple MR constructs in use in 2009, and it isn't clear to me if that framework can be used here. That power paper suggests that weak genetic instruments need very large sample sizes, far larger than what is used in the current manuscript. I am unable to estimate the true strength of the instruments used here, and so I am unsure of whether power is an issue or not.

We have now included power calculations in our manuscript to address the reviewer’s concerns. Nevertheless, as mentioned above, post-hoc power calculations are of limited value, as statistical power is already reflected in the uncertainty around the point estimates (the 95% confidence intervals). Hence, it is important to avoid drawing conclusions regarding the likelihood of true effects or false negatives based on these calculations.

Although the hypothesis here is that smoking accounts for the apparent BMI association previously reported for HNC, it would have been preferable to see the estimates for their 2 genetic instruments for tobacco alone. The current results only show the BMI instruments alone and then with the tobacco instruments. I would like to see what the risk estimates are for the tobacco instrument alone, so that I can judge for myself what happens in the joint models. As presented, one can only do that for the BMI instruments.

We thank the reviewer for this comment. The univariable IVW MR estimate of smoking initiation was OR=2.07 (95%CI 1.60 to 2.68, p<0.001), while the one for comprehensive smoking index was OR=4.47 (95%CI 3.31 to 6.03, p<0.001). We have included this information in the manuscript as requested (please see response to reviewing editor above).

On line 319, they write that "We did not find evidence against bias due to correlated pleiotropy..." I find this difficult to parse, but I think it means that they should believe that correlated pleiotropy remains a problem. So again, they seem to see their primary model as compromised, and so do I. This limitation is again stated by the authors on lines 351-352.

We apologise if the wording of the sentence was not easy to understand. When using the CAUSE method, we did not find evidence to reject the null hypothesis that the sharing (correlated pleiotropy) model fits the data at least as well as the causal model. In other words, our CAUSE finding and the inconsistencies observed across our other sensitivity analyses led us to believe that our main IVW MR estimate for BMI-HNC was likely biased by correlated pleiotropy. We believe it is important to explore the source of this bias, which is why we used multivariable MR to investigate the direct effect of BMI on HNC risk while accounting for smoking behaviour.

In the following paragraphs (lines 358-369), the authors state that their findings are consistent with prior reports, but that doesn't seem to be the case if we take their primary BMI instrument as representing the outcome of this manuscript. Here, they find an association between the BMI instrument and HNC risk, but in each of the other papers they present the primary finding was null without the extensive model changes or the aim of accounting for tobacco with another instrument. I don't see that as replication.

This is a good point. We have now edited the discussion of our manuscript to avoid giving the impression that our findings replicate those from studies that do not account for smoking behaviour in their analyses. We have edited lines 384-401 as follows:

“Previous MR studies suggest adiposity does not influence HNC risk[27-29]. Gormley et al.[28] did not find a genetically predicted effect of adiposity on combined oral and oropharyngeal cancer when investigating either BMI (OR=0.89 per 1-SD, 95% CI 0.72–1.09, p=0.26), WHR (OR=0.98 per 1-SD, 95% CI 0.74–1.29, p=0.88) or waist circumference (OR=0.73 per 1-SD, 95% CI 0.52–1.02, p=0.07) as risk factors. Similarly, a large two-sample MR study by Vithayathil et al.[29] including 367,561 UK Biobank participants (of which 1,983 were HNC cases) found no link between BMI and HNC risk (OR=0.98 per 1-SD higher BMI, 95% CI 0.93–1.02, p=0.35). Larsson et al.[27] meta-analysed Vithayathil et al.’s[29] findings with results obtained using FinnGen data to increase the sample size even further (N=586,353, including 2,109 cases), but still did not find a genetically predicted effect of BMI on HNC risk (OR=0.96 per 1-SD higher BMI, 95% CI 0.77–1.19, p=0.69). With a much larger sample (N=31,523, including 12,264 cases), our IVW MR analysis suggested BMI may play a role in HNC risk, in contrast to previous studies. However, our sensitivity analyses implied that causality was uncertain.”

We also deleted part of a sentence in the discussion section, so lines 416-418 now look as follows: “An important strength of our study was that the HEADSpAcE consortium GWAS used had a large sample size which conferred more statistical power to detect effects of adiposity on HNC risk compared to previous MR analyses[27-29].”

On lines 384-386 they note a strength is that this is the largest study to date, but I would reiterate that larger and more powerful does not equate to adequately powered.

This is true. We have included power calculations in the manuscript as requested.

It's well known that different HNC subsites have different etiologies, as they mention on lines 391-392, and it is implicit in their use of data on HPV positive and negative oropharyngeal cancer. They say that they did not find evidence for heterogeneity in this study, but that would only be true for the null BMI instrument. The effect sizes for their smoking instruments are strikingly different between the subsites.

We agree and are sorry for the confusion we may have caused by the way we worded our findings. We have edited the text to clarify that the lack of subsite heterogeneity only applied to our results for BMI/WHC/WC-HNC risk. Lines 418-424 now read as follows:

“Furthermore, the availability of data on more HNC subsites, including oropharyngeal cancers by HPV status, allowed us to investigate the relationship between adiposity and HNC risk in more detail than previous MR studies which limited their subsite analyses to oral cavity and overall oropharyngeal cancers[28, 68]. This is relevant because distinct HNC subsites are known to have different aetiologies[69], although we did not find evidence of heterogeneity across subsites in our analyses investigating the genetically predicted effects of BMI, WHR and WC on HNC risk.”

Finally, the literature on mutational patterns gives us strong reason to believe that HNC caused by tobacco are biologically distinct from tumors not caused by tobacco. The authors report in the introduction that traditional observational studies of BMI and HNC have reported different findings in smokers versus never smokers, so I would assume there is a possibility that the BMI instrument could have different associations with tumors of the tobacco-induced phenotype and tumors with a non-tobacco induced phenotype. I would assume that authors have access to the data on self-reported tobacco use behavior, even if they can't separate these tumors by molecular types. Stratifying their analysis by tobacco users or not might reveal different results with the BMI instrument.

We appreciate the reviewer’s comment. We agree that it would have been interesting to present stratified analyses by smoking status along our main findings. However, we decided against this because of the risk of inducing collider bias in our MR analyses i.e., where stratifying on smoking status may induce spurious associations between the adiposity instruments and confounding factors. Multivariable MR is considered a better way of investigating the direct effects of an exposure (adiposity) on an outcome (HNC) accounting for a third variable (smoking)[14], which is why we opted for this method instead.

References:

(1) Heinsberg LW, Weeks DE: Post hoc power is not informative. Genet Epidemiol 2022, 46(7):390-394.

(2) Burgess S, Butterworth A, Thompson SG: Mendelian randomization analysis with multiple genetic variants using summarized data. Genet Epidemiol 2013, 37(7):658-665.

(3) Burgess S, Davey Smith G, Davies NM, Dudbridge F, Gill D, Glymour MM, Hartwig FP, Kutalik Z, Holmes MV, Minelli C et al: Guidelines for performing Mendelian randomization investigations: update for summer 2023. Wellcome Open Res 2019, 4:186.

(4) Morris RW, Taylor AE, Fluharty ME, Bjorngaard JH, Asvold BO, Elvestad Gabrielsen M, Campbell A, Marioni R, Kumari M, Korhonen T et al: Heavier smoking may lead to a relative increase in waist circumference: evidence for a causal relationship from a Mendelian randomisation meta-analysis. The CARTA consortium. BMJ Open 2015, 5(8):e008808.

(5) Taylor AE, Morris RW, Fluharty ME, Bjorngaard JH, Asvold BO, Gabrielsen ME, Campbell A, Marioni R, Kumari M, Hallfors J et al: Stratification by smoking status reveals an association of CHRNA5-A3-B4 genotype with body mass index in never smokers. PLoS Genet 2014, 10(12):e1004799.

(6) Taylor AE, Richmond RC, Palviainen T, Loukola A, Wootton RE, Kaprio J, Relton CL, Davey Smith G, Munafo MR: The effect of body mass index on smoking behaviour and nicotine metabolism: a Mendelian randomization study. Hum Mol Genet 2019, 28(8):1322-1330.

(7) Asvold BO, Bjorngaard JH, Carslake D, Gabrielsen ME, Skorpen F, Smith GD, Romundstad PR: Causal associations of tobacco smoking with cardiovascular risk factors: a Mendelian randomization analysis of the HUNT Study in Norway. Int J Epidemiol 2014, 43(5):1458-1470.

(8) Carreras-Torres R, Johansson M, Haycock PC, Relton CL, Davey Smith G, Brennan P, Martin RM: Role of obesity in smoking behaviour: Mendelian randomisation study in UK Biobank. BMJ 2018, 361:k1767.

(9) Freathy RM, Kazeem GR, Morris RW, Johnson PC, Paternoster L, Ebrahim S, Hattersley AT, Hill A, Hingorani AD, Holst C et al: Genetic variation at CHRNA5-CHRNA3-CHRNB4 interacts with smoking status to influence body mass index. Int J Epidemiol 2011, 40(6):1617-1628.

(10) Thorgeirsson TE, Gudbjartsson DF, Sulem P, Besenbacher S, Styrkarsdottir U, Thorleifsson G, Walters GB, Consortium TAG, Oxford GSKC, consortium E et al: A common biological basis of obesity and nicotine addiction. Transl Psychiatry 2013, 3(10):e308.

(11) Wills AG, Hopfer C: Phenotypic and genetic relationship between BMI and cigarette smoking in a sample of UK adults. Addict Behav 2019, 89:98-103.

(12) Coscia C, Gill D, Benitez R, Perez T, Malats N, Burgess S: Avoiding collider bias in Mendelian randomization when performing stratified analyses. Eur J Epidemiol 2022, 37(7):671-682.

(13) Hamilton FW, Hughes DA, Lu T, Kutalik Z, Gkatzionis A, Tilling K, Hartwig FP, Davey Smith G: Non-linear Mendelian randomization: evaluation of effect modification in the residual and doubly-ranked methods with simulated and empirical examples. Eur J Epidemiol 2025.

(14) Sanderson E, Davey Smith G, Windmeijer F, Bowden J: An examination of multivariable Mendelian randomization in the single-sample and two-sample summary data settings. Int J Epidemiol 2019, 48(3):713-727.